# Nested mechanosensory feedback actively damps visually guided head movements in *Drosophila*

**Benjamin Cellini, Jean-Michel Mongeau\***

Department of Mechanical Engineering, Pennsylvania State University, University Park, United States

**Abstract** Executing agile locomotion requires animals to integrate sensory feedback, often from multiple sources. For example, human gaze is mediated by multiple feedback loops that integrate visual and vestibular information. A central challenge in studying biological feedback loops is that they are nested and dynamically coupled. Here, we develop a framework based on control theory for unraveling nested feedback systems and apply it to study gaze stabilization in the fruit fly (*Drosophila*). By combining experimental and mathematical methods to manipulate control topologies, we uncovered the role of body-generated mechanosensory feedback nested within visual feedback in the control of head movements. We discovered that visual feedback changed the tuning of head movements across visual motion frequencies whereas mechanosensory feedback damped head movements. Head saccades had slower dynamics when the body was free to move, further pointing to the role of damping via mechanosensory feedback. By comparing head responses between self-generated and externally generated body motion, we revealed a nonlinear gating of mechanosensory feedback that is motor-context dependent. Altogether, our findings reveal the role of nested feedback loops in flies and uncover mechanisms that reconcile differences in head kinematics between body-free and body-fixed flies. Our framework is generalizable to biological and robotic systems relying on nested feedback control for guiding locomotion.

**\*For correspondence:**
jmmongeau@psu.edu

**Competing interest:** The authors declare that no competing interests exist.

## Editor's evaluation

The manuscript makes an important contribution to feedback control in neural systems. The analysis and modeling together make a compelling case for a nested system, combining visual with mechanosensory feedback, for head and body control in the fruit fly. The experiments that support these results are compelling and well-executed and the strategies for dissecting and modeling feedback are valuable to the field, and broadly applicable to other neural control systems. This paper will reach a wide audience; researchers investigating biological control systems, visual feedback, and gaze stabilization will all be interested in these results.

## Introduction

Animal locomotion and the associated neural computations are closed-loop (*Cowan et al., 2014*; *Roth et al., 2014*; *Madhav and Cowan, 2020*). During locomotion, sensory systems measure external and internal states. This sensory information is then processed by the brain to guide motor decisions and the resulting movement shapes sensory inputs, thus closing the loop. Visually active animals often integrate visual and mechanosensory information to guide movement through complex environments (*Mongeau et al., 2021*; *Frye, 2010*). For instance, hawk moths integrate information from visual and mechanosensory pathways when feeding from flowers moving in the wind (*Roth et al., 2016*), glass

**Figure 1.** Parallel and nested sensory fusion in biological systems. (**A**) Control model of parallel sensory fusion. Multiple sensory systems, $S_1$ and $S_2$, measure an external reference state $R$ with respect to the system's motion $Y$. The information measured by $S_1$ and $S_2$ is fused together in parallel by a neural controller $C$ to maintain equilibrium. The neural controller drives locomotion through the system's biomechanics $P$, which feeds back to shape future sensory inputs, thus closing the loop. (**B**) Control model of nested sensory fusion. Same as (A) but one of the sensory systems ($S_2$) does not directly measure the external reference state $R$. Instead the system state is directly fed to the neural controller $C$ (purple). Thus $S_2$ is not involved with measuring external sensory states.

knifefish rely on a combination of visual and electrosensory feedback to regulate their position within a moving refuge (*Sutton et al., 2016*), and flies stabilize vision via antennal feedback (*Fuller et al., 2014b*). For many of these behaviors, the sensors—e.g. eyes and proboscis in hawk moths—measure the same information (e.g. flower motion) in parallel. This is fundamentally a parallel sensory fusion problem where the animal must weight information from parallel pathways (*Figure 1A*). The Kalman filter has been applied to biological and robotic systems to solve similar sensory fusion problems and determine the optimal weight for each sensor (*Sun and Deng, 2004*; *Ernst and Banks, 2002*).

In contrast to parallel sensory fusion—where sensory information operates at the same level in the control hierarchy—sensory feedback is often nested within higher levels of control (*Figure 1B*, *Hardcastle and Krapp, 2016*; *Mongeau et al., 2021*). Consider the goal-directed task of visually navigating through a complex environment. Vision provides slower and higher level information for guidance whereas mechanosensory inputs due to self-motion—measured by the vestibular and somatosensory systems—influence rapid, lower level postural reflexes (*Bent et al., 2004*; *Goldberg et al., 2012*; *Nakahira et al., 2021*). In this context, mechanosensory feedback is *nested* within visual feedback because it is activated by visually guided locomotion, and thus it does not directly relate to the task goal (*Mongeau et al., 2021*). This sensorimotor organization is analogous to cascade control in engineering controller design, where there are inner feedback loops nested within outer loops (*Krishnaswamy et al., 1990*). While many prior studies have investigated how animals integrate sensory information from multiple pathways, the case where one sensory pathway is nested within another has received significantly less attention. How does the brain integrate nested sensory feedback for effective locomotion?

One exemplar sensorimotor system that includes nested mechanosensory feedback is the gaze stabilization reflex. Primates move their eyes and head in response to visual motion to stabilize gaze,

termed the optokinetic response (OKR) (*Land, 2019*). Eye and head movements feedback to shape visual inputs, and measurements of head motion from the vestibular system feedback to keep the eyes steady with respect to the head—termed the vestibulo-ocular reflex (VOR) (*Goldberg et al., 2012*). Although both the OKR and the VOR are reflexive stabilization feedback loops, the VOR is nested within the OKR. Prior work showed that the OKR and VOR are inversely tuned: the OKR responds strongly to low frequencies and the VOR is tuned to higher frequencies (*Schweigart et al., 1997*; *Barnes, 1993*). However, the specific contributions of visual and nested mechanosensory feedback when the OKR and VOR are active together remains unclear. Intriguingly, the gaze stabilization

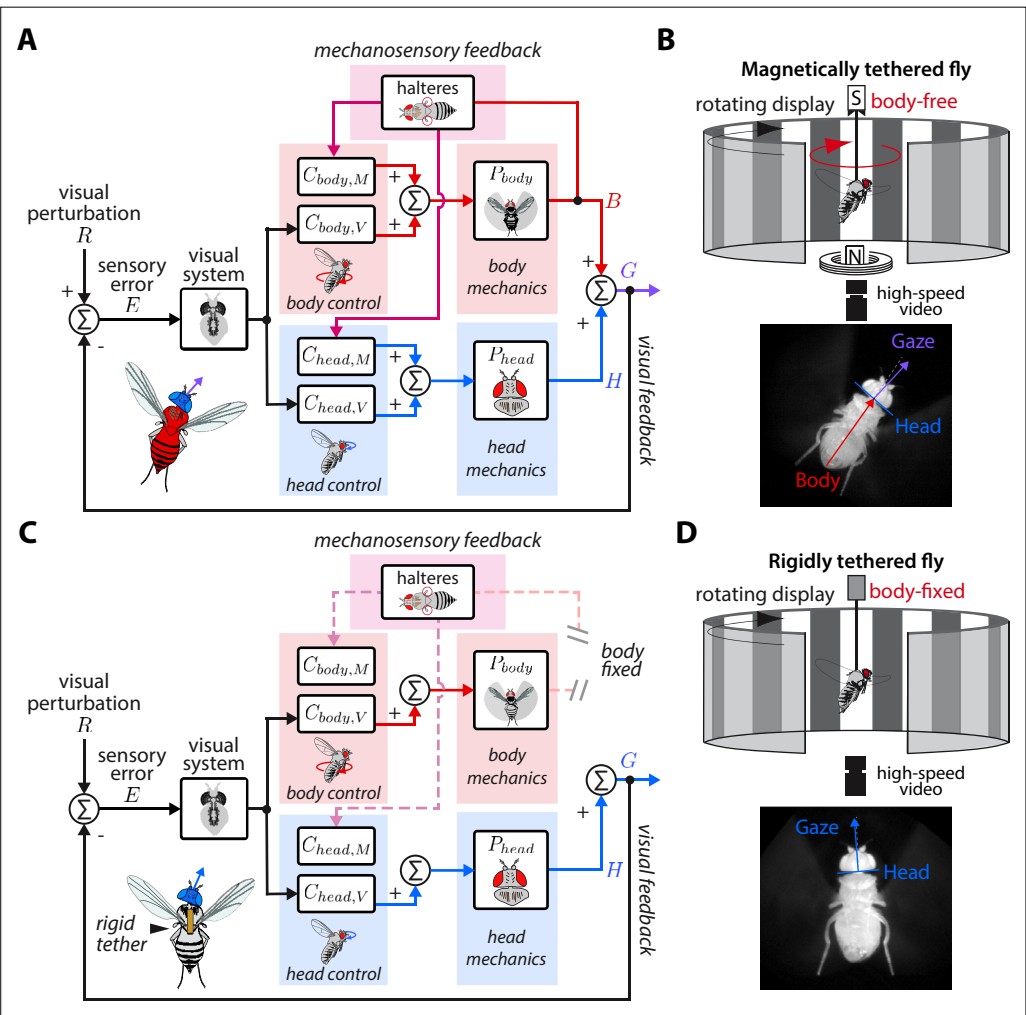

**Figure 2.** Control model of visual and nested mechanosensory feedback during gaze stabilization in fly flight. (**A**) The control framework used to model and analyze the gaze stabilization system in body-free flies. Flies respond to an external visual perturbation $R$ by attempting to minimize the sensory visual error $E$ measured by their visual system. Neural control circuits in the brain for the head $C_{head,V}$ and body $C_{body,V}$ process the sensory error and send motor control signals to the corresponding biomechanical systems $P_{head}$ and $P_{body}$ to generate head $H$ and body $B$ movements. The fly's gaze $G$ is controlled by the sum of head and body movements, which feeds back to shape the sensory error entering the visual system. Flies also measure mechanosensory information associated with body motion via the halteres, which is processed in the brain by analogous controllers for the head $C_{head,M}$ and body $C_{body,M}$, and also contributes to shaping head and body responses. In this paradigm, mechanosensory feedback is nested within visual feedback. (**B**) The magnetic tether experimental paradigm for body-free flies corresponding to (**A**). A fly is tethered to a pin which is placed in a low-friction bearing and suspended in a magnetic field, allowing free rotation about the vertical (yaw) axis. (**C**) Same as (**A**) but for a body-fixed fly. Note that contributions of body visual feedback and nested mechanosensory feedback due to body motion are no longer present. The fly's gaze is now purely determined by head movements. (**D**) The rigid tether experimental paradigm for body-fixed flies corresponding to (**C**).

response of flies shows close parallels to the primate visuomotor response, with similar feedback topology, making it an accessible model system to unravel the mechanisms underlying nested feedback control (*Cellini et al., 2022*; *Elzinga et al., 2012*).

Here, we studied nested sensorimotor feedback loops in fruit flies (*Drosophila*), with a specific focus on teasing apart the contributions of visual and nested mechanosensory feedback during gaze stabilization. The gaze stabilization reflex consists of multiple motor systems—the head and body—that operate in closed-loop with the goal of reducing optic flow across the retina and keeping gaze level (*Figure 2A*, *Cellini et al., 2022*; *Cellini and Mongeau, 2020a*). The halteres—gyroscope-like organs that encode body velocity by sensing gyroscopic forces and structure the timing of motor outputs in flies (*Fraenkel, 1939*; *Nalbach and Hengstenberg, 1994*; *Dickerson et al., 2019*)—also influence the control of head and body movements about all three rotational axes (*Hardcastle and Krapp, 2016*; *Mureli and Fox, 2015*; *Mureli et al., 2017*; *Nalbach, 1993*; *Rauscher and Fox, 2021*). When visual inputs activate the gaze stabilization reflex and drive a compensatory response of the head and body, the halteres presumably sense the resulting body velocity and provide mechanosensory information that further influences a fly's visuomotor behavior. Thus it would follow that mechanosensory feedback is inherently nested within visual feedback. Studying gaze stabilization in flies can therefore provide insights into how nested feedback loops interconnect and shape higher level loops in animal locomotion. Established experimental paradigms for studying fly flight provide a unique opportunity to manipulate control topologies, allowing us to break feedback loops and tease out the role of visual and nested mechanosensory feedback. In contrast to prior work that studied the parallel integration of visual and haltere information in open-loop—where flies had their head and body fixed in place (*Dickinson, 1999*; *Sherman and Dickinson, 2003*; *Sherman and Dickinson, 2004*)—here we employ an experimental paradigm that allowed flies to freely move their head and body in closed-loop (*Figure 2B*). In conjunction with empirical data, we synthesized a control model for mathematically teasing apart the role of nested sensory feedback. We applied this model to study how body-generated visual and nested mechanosensory feedback are integrated during the control of head movements. Our results provide new insights into how nested sensory feedback may be structured across phyla for gaze stabilization.

## Results

### Control model of visual and nested mechanosensory feedback during gaze stabilization in fly flight

During flight, flies are often knocked off course by gusts of wind or other external perturbations to their flight paths. Such perturbations ($R$) generate optic flow relative to a fly's own motion, or sensory error ($E$), across the retina that is processed by the brain to generate corrective steering maneuvers of the head ($H$) and/or body ($B$)—with the goal to minimize $E$ (*Figure 2A*, *Cellini et al., 2022*; *Cellini and Mongeau, 2020a*; *Dickinson and Muijres, 2016*). Mechanosensory information from externally-generated and/or self-generated body motion—and measured by the halteres—also elicits corrective movements of the head and wings/body (*Figure 2A*, *Hardcastle and Krapp, 2016*; *Dickinson, 1999*; *Sherman and Dickinson, 2003*; *Hengstenberg, 1988*; *Sandeman, 1980*; *Beatus et al., 2015*). This suite of multisensory reflexes keeps gaze level and appears essential for flight.

We developed a control model of gaze stabilization about the vertical (yaw) axis to model the flow of visual and mechanosensory information in driving head and body motor responses in flies (*Figure 2A*). We based our framework on prior models in flies that demonstrated that inputs from the visual system and halteres sum in the nervous system (*Sherman and Dickinson, 2004*), but go further by including naturalistic closed-loop feedback and mechanics. For the head and body, we modeled the distinct contributions of sensory feedback by separating the neural control of gaze stabilization into two sub-components, one for visual feedback ($C_{head,V}$ and $C_{body,V}$) and the other for mechanosensory feedback ($C_{head,M}$ and $C_{body,M}$). Critically, we assumed that visual feedback has a gain of –1—because motion in one direction generates equal and opposite optic flow—and that that the mechanosensory neural controllers ($C_{head,M}$ and $C_{body,M}$) only receive haltere inputs *due to body motion*. The other functions of the halteres related to structuring the timing of motor output are assumed to be contained within the dynamics of the visual controllers (*Dickerson et al., 2019*). The separate neural controller pairs for the head and body in our model ensured that any differences in tuning between the head and body were

considered. We assumed approximately linear time-invariant (LTI) dynamics (*Aström et al., 2010*), which is supported by experimental data from prior work (*Cellini and Mongeau, 2020a*; *Cellini et al., 2022*). Finally, our model only considered mechanosensory feedback generated from self-generated body motion, asserting that mechanosensory feedback is nested within visual feedback (*Figure 2A*).

We first modeled the head response by defining the transforms mapping visual and mechanosensory inputs to head motor responses:

$$G_{head,V} = P_{head} \, C_{head,V} \tag{1}$$

$$G_{head,M} = P_{head} \, C_{head,M}. \tag{2}$$

$G_{head,V}$ represents the visual transform from $E$ to $H$, and $G_{head,M}$ represents the mechanosensory transform from $B$ to $H$. These transforms consist of the multiplication of the corresponding neural circuits associated with the visual ($C_{head,V}$) and mechanosensory ($C_{head,M}$) control centers of the fly brain and the passive biomechanics of the head-neck system ($P_{head}$). We assume that the dynamics of the sensory systems—visual system and halteres—are contained within the dynamics of $C_{head,V}$ and $C_{head,M}$, respectively. All the transforms and signals in our model are designated as non-parametric complex-valued functions (see Materials and methods). Throughout, we omit the complex variable $s$ for brevity.

Using the transforms *Equation 1* and *Equation 2*, we derived an expression for the closed-loop head motor response $H$ as a function of an external visual perturbation $R$ and body motion $B$ (see Materials and methods for more detailed derivations). We first defined the head response as the sum of visual and mechanosensory inputs due to body motion:

$$H = G_{head,V}E + G_{head,M}B, \tag{3}$$

where the sensory error $E$ is equivalent to the visual perturbation $R$ subtracted by the fly's gaze (sum of $H$ and $B$):

$$E = R - H - B. \tag{4}$$

Substituting *Equation 4* into *Equation 3* and solving for $H$ yields the expression for the closed-loop head response:

$$H = \underbrace{\frac{G_{head,V}}{1 + G_{head,V}}R}_{\substack{\text{head visual} \\ \text{feedback}}} - \underbrace{\frac{G_{head,V}}{1 + G_{head,V}}B}_{\substack{\text{body visual} \\ \text{feedback}}} + \underbrace{\frac{G_{head,M}}{1 + G_{head,V}}B}_{\substack{\text{body mechanosensory} \\ \text{feedback}}}. \tag{5}$$

Notably, the closed-loop head response of body-free flies (*Equation 5*) is mediated by three sources of sensory feedback: (1) visual feedback from head movements themselves, (2) visual feedback from body motion and (3) nested mechanosensory feedback from body motion. Conversely, the head response of body-fixed flies can be represented as:

$$H = \underbrace{\frac{G_{head,V}}{1 + G_{head,V}}R}_{\substack{\text{head visual} \\ \text{feedback}}} \tag{6}$$

where all terms associated with body motion are set to zero ($B = 0$ in *Equation 5*), leaving only visual feedback from head movements (*Figure 2C*). We recognize that haltere neural inputs are always present—even when there is no body motion—and can not be completely abolished without removing the halteres. However these inputs are involved with structuring the timing of motor output, not with the encoding of body velocity via sensing gyroscopic forces (*Dickerson et al., 2019*; *Fayyazuddin and Dickinson, 1996*; *Fayyazuddin and Dickinson, 1999*). Thus, we lump this tonic function of the halteres into the visual controllers ($C_{head,V}$ and $C_{head,B}$), which are present in both body-free and body-fixed flies. Crucially, our control model mathematically predicts that the head motor responses of body-free (*Equation 5*) and body-fixed (*Equation 6*) flies will be distinct due to differences in sensory feedback. Therefore, comparing how the head responses of body-free and body-fixed flies differ provides insights into the distinct sensory modalities that influence head control.

## Sensory feedback generated from body movements alters the magnitude, timing, and performance of head responses

Our control model predicted that body-free and body-fixed flies should have distinct head motor responses to the same visual perturbation due to the absence of body-generated visual and mechanosensory feedback. To determine whether empirical data supported this prediction, we employed two experimental paradigms: (1) a magnetic tether where flies were tethered to a pin and suspended between two magnets, allowing free, closed-loop body rotation about the vertical (yaw) axis (*Figure 2B*) and (2) a rigid tether where flies were fixed in place, thus opening body-generated visual and mechanosensory feedback loops (*Figure 2D*).

We presented flies in both paradigms with visual perturbations consisting of single sinusoids with frequencies spanning 0.7–10.6 Hz to reveal how body-generated feedback influences head motor responses (see Materials and methods). We began by quantifying the body response of body-free flies to understand what frequency range body-generated feedback should have the most impact on head responses. The body was strongly tuned to low frequencies, similar to a low-pass filter (*Figure 3A*, *red*, *Figure 3—video 1*–*Figure 3—video 2*, *Cellini et al., 2022*). Thus, body visual feedback reduced the optic flow—or sensory error ($E$)—entering the visual system, at low frequencies especially (*Figure 3A–B*, *yellow*, *Figure 4A*, *red*). This result, combined with our control model, predicted that any differences between head responses in body-free and body-fixed flies should be the greatest at low frequencies. Because biological systems often exhibit nonlinear behavior to different types of sensory inputs, we also measured fly responses to sum-of-sines visual perturbations as a check for linearity. Although flies responses varied slightly between single-sine and sum-of-sine perturbations, the overall behavior was similar, indicating that our findings are generalizable (*Figure 3—figure supplement 1A-B*, *Figure 3—video 4*).

Next, we quantified the head responses of body-free and body-fixed flies. Consistent with prior work, body-free flies generated head movements that were inversely tuned to the body and resembled a high-pass filter, where the head operated with the largest gains at high frequencies (*Figure 3C*, *blue*) (*Cellini et al., 2022*). Consistent with the prediction of our model (*Equation 6*), the head response of body-fixed flies was appreciably different from that of body-free flies (*Figure 3C*, *Figure 3—figure supplement 1A-B*, *Figure 3—video 1*–*Figure 3—video 2*, *Cellini and Mongeau, 2020a*). Specifically, body-fixed flies moved their head with larger magnitude than body-free flies (*Figure 3C*). Interestingly, head movements in body-fixed flies were often driven to the anatomical limits of the neck joint (approximately ±15°), which was never the case for body-free flies (*Figure 3C*). This led to head trajectories that were saturated, meaning that the head could possibly have moved with larger amplitude if it were anatomically possible. The total distributions of head angular displacements were likewise significantly different between body-free and body-fixed flies (F-test, $p < 0.001$ for every frequency) (*Figure 3D*, *Figure 3—figure supplement 1C*). Body-free flies rarely moved their head more than 5° from the neutral position (0°), whereas body-fixed flies regularly moved their head in excess of 10° (*Figure 3D*, *Figure 3—figure supplement 1C*). This was especially prominent at lower frequencies, while head responses at higher frequencies were closer in magnitude (*Figure 3C*, *Figure 3—figure supplement 1A, B*). This is consistent with the low-frequency tuning of body movements, where the smaller sensory error in body-free flies at low frequencies likely led to the smaller head motor responses. Body visual feedback also altered the phase, or the timing, of the sensory error signal entering the visual system, leading to differences in the timing of head motor responses in body-free and body-fixed flies (*Figure 3A, C*).

To quantify the performance of head responses, we measured the closed-loop transforms from $R$ to $H$ in body-free and body-fixed flies. These transforms mirror *Equation 5* and *Equation 6*, respectively, but normalize the head response with respect to $R$. While these transforms are complex valued functions, we represented them graphically via gain, phase, and compensation error (*Figure 4B*). Gain represents the magnitude of the ratio of the perturbation and head $|\frac{H}{R}|$ phase represents the difference in timing $\angle\frac{H}{R}$, and compensation error describes the normalized magnitude of the sensory error signal $|\frac{E}{R}|$. A compensation error value of zero indicates ideal performance, a value between zero and one indicates intermediate performance, a value of one indicates that the head response has no effect on performance, and a value greater than one indicates a deleterious response (see Materials and methods). Body-fixed flies operated with overall higher gain than body-free flies (*Figure 4B*, *Figure 4—figure supplement 1A*, *purple* vs *blue*). Both body-free and body-fixed flies displayed a

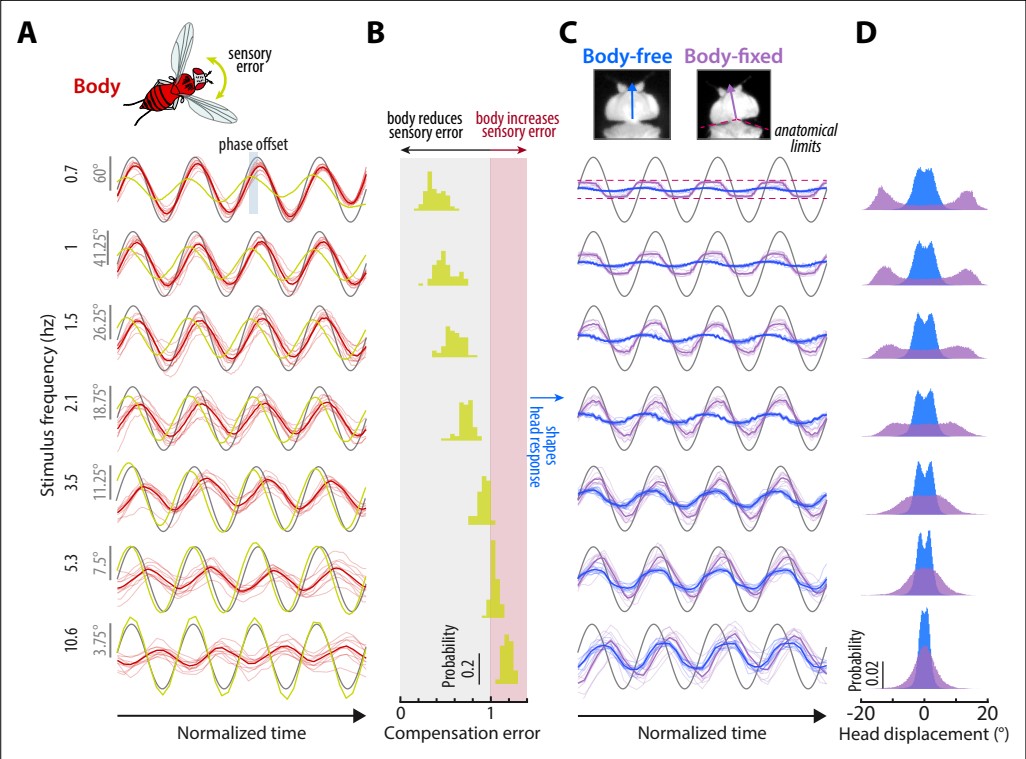

**Figure 3.** Sensory feedback generated from body movements alters the magnitude, timing, and performance of head responses. (**A**) The body response (red) of body-free flies to single-sine visual perturbations (grey) with varying frequency. The x-axis is normalized to show four oscillations at each frequency. Note that the body response is larger relative to the visual perturbation at low frequencies, leading to a smaller sensory error signal (yellow) in the head reference frame. Thick lines: mean. Thin lines: individual fly means. (**B**) The distribution of compensation errors in the head reference frame corresponding to the sensory error in (A) normalized by the perturbation amplitude. Values below one indicate that body movements reduced the sensory error while values greater than one indicate that body movements increased the sensory error. (**C**) The head response of body-free (blue) and body-fixed (violet) flies to the same visual perturbation (grey) shown in (A). At low frequencies, the head would often run into the anatomical limits of the neck joint (dashed pink lines). Thick lines: mean response. Thin lines: individual fly means. (**D**) The total distribution of head angular displacements for body-free and body-fixed flies for each perturbation frequency. For each frequency, the body-free and body-fixed head distributions had a different variance (F-test, $p < 0.001$). Body-free: $n = 10$ flies, Body-fixed: $n = 9$ flies.

The online version of this article includes the following video and figure supplement(s) for figure 3:

**Figure supplement 1.** Sum-of-sines body and head responses.

**Figure supplement 2.** Saturation-corrected head response and LSSA sensitivity analysis.

**Figure 3—video 1.** Comparison of a body-free fly (magnetic tether, left) and body-fixed fly (rigid tether, right) head response to a 1 Hz sine wave visual perturbation.
https://elifesciences.org/articles/80880/figures#fig3video1

**Figure 3—video 2.** Same as *Figure 3—video 1* but for a 2.1 Hz visual perturbation.
https://elifesciences.org/articles/80880/figures#fig3video2

**Figure 3—video 3.** Same as *Figure 3—video 1* but for a 5.3 Hz visual perturbation.
https://elifesciences.org/articles/80880/figures#fig3video3

**Figure 3—video 4.** Same as *Figure 3—video 1* but for a sum-of-sines visual perturbation.
https://elifesciences.org/articles/80880/figures#fig3video4

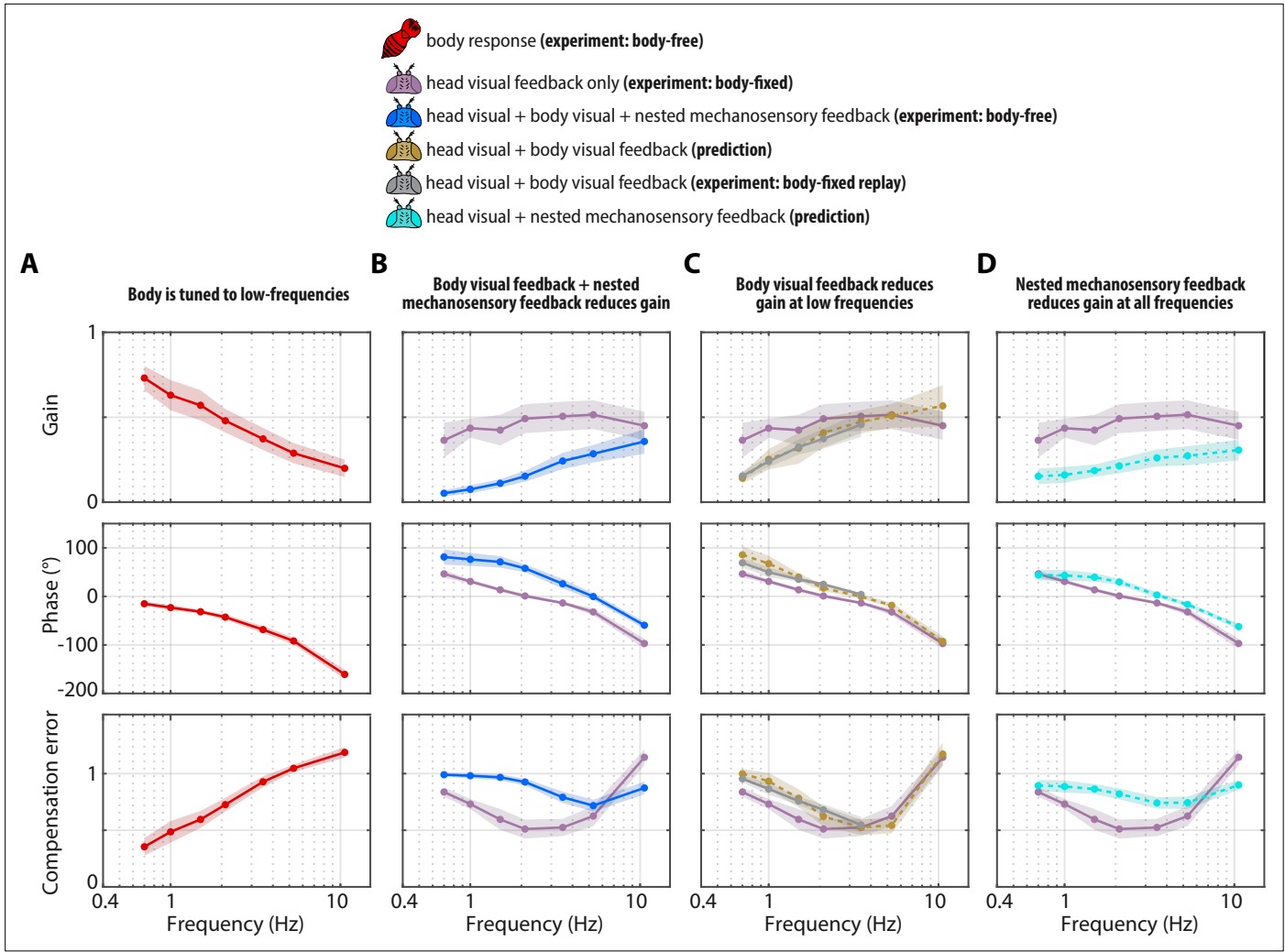

**Figure 4.** Visual and mechanosensory feedback mediate head control. The legend indicates whether the data corresponds to the head or body, , the sources of sensory feedback, and the relevant experiment (or prediction). (**A**) The closed-loop transform from the visual perturbation $R$ to the body response $B$. Note that the body is primarily tuned to low frequencies. (**B**) The closed-loop transform from the visual perturbation $R$ to the head response $H$ for different sensory feedback conditions. The head transform measured in body-free flies (blue) contains all three sources of feedback (see *Equation 3*), while the head transform measured in body-fixed flies (purple) contains only head visual feedback (see *Equation 6*). (**C**) The predicted (dashed line) transform for the head response with head and body visual feedback (copper, see *Equation 7*, corresponding to *Figure 4—figure supplement 2A*) and the experimentally measured equivalent from a 'replay' experiment (grey, corresponding to *Figure 4—figure supplement 2C-E*). The highest two frequencies were omitted in the replay experiment due to limitations of our flight arena display system (see Materials and methods). (**D**) The predicted (dashed line) transform for the response with head visual feedback and body mechanosensory feedback (cyan, see *Equation 8*, corresponding to *Figure 4—figure supplement 2B*). Body-free: $n = 10$ flies, Body-fixed: $n = 9$ flies, Body-fixed replay: $n = 4$ flies. For all panels, shaded regions: ±1 STD. Also see *Figure 4—figure supplement 3* for all plots overlaid to facilitate comparison across groups.

The online version of this article includes the following figure supplement(s) for figure 4:

**Figure supplement 1.** Sum-of-sines transforms.

**Figure supplement 2.** Control diagrams and replay experiment for manipulations of sensory feedback.

**Figure supplement 3.** Overlaid transforms.

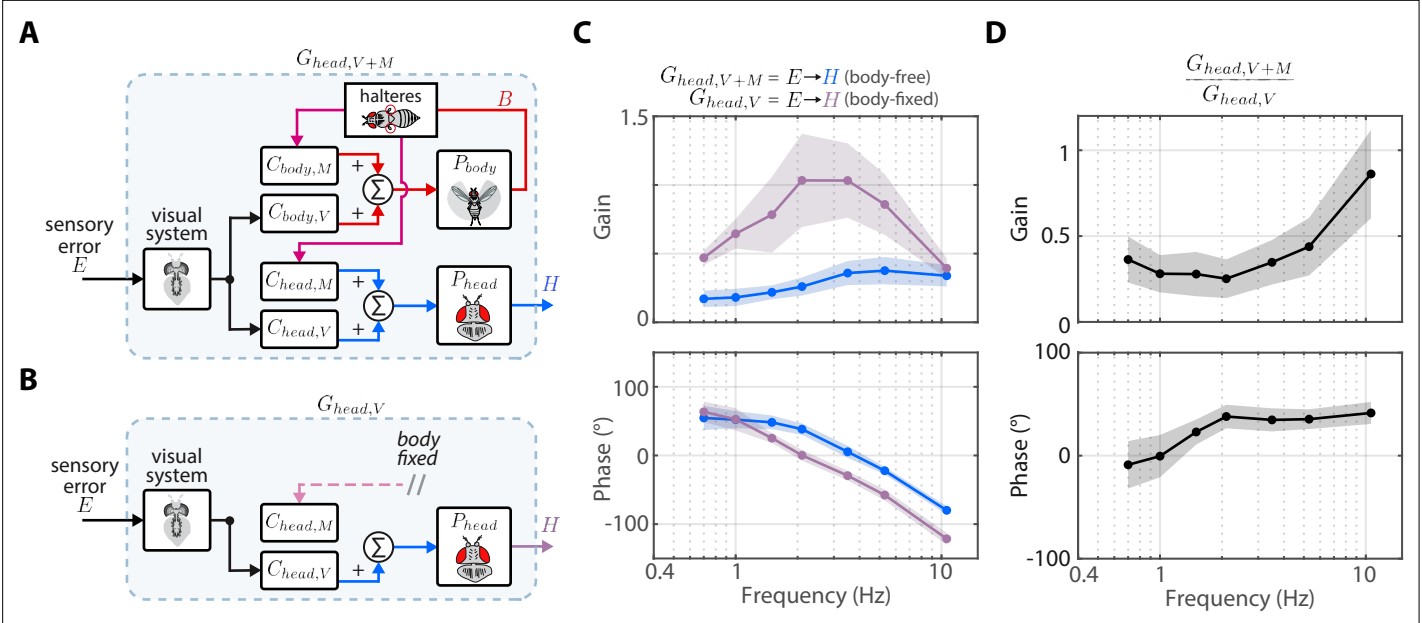

**Figure 5.** Nested mechanosensory feedback damps head movements. (**A**) The control diagram of the sensory error $E$ to head $H$ transform $G_{head,V+M}$ in body-free flies. Note that this transform includes nested mechanosensory feedback from body motion. (**B**) The control diagram of the sensory error $E$ to head $H$ transform in body-fixed flies, which is simply the visual transform $G_{head,V}$. (**C**) The gain and phase of the $E$ to $H$ transform for body-free (blue) and body-fixed (purple) flies. Shaded regions: ±1 STD (**D**) The ratio of the $E$ to $H$ transform in body-free and body-fixed flies ($G_{head,V+M}/G_{head,V}$). If nested mechanosensory feedback from body motion had no effect, we would expect this ratio to have a gain of one and phase of 0 (dashed blue lines). The empirical data has a gain less than one, indicating the head movements are damped by nested mechanosensory feedback. Shaded regions: ±1 STD. Body-free: $n = 10$ flies, Body-fixed: $n = 9$ flies.

phase lead (phase >0) at low-frequencies which decreased with increasing frequency, however this was less pronounced in body-fixed flies (*Figure 4B*, *Figure 4—figure supplement 1A*). The larger gain and smaller phase lead in body-fixed flies led to improved performance at low-frequencies, but worse performance at higher frequencies, illustrating that body-fixation leads to tradeoffs in head stabilization performance (*Figure 4A*, *Figure 4—figure supplement 1A*, *compensation error*). Altogether, the magnitude, timing, and performance of head motor responses were distinct between body-fixed and body-free flies, demonstrating the critical role of sensory feedback in shaping head movements.

## Visual feedback changes the tuning of head responses across visual motion frequencies

Body-free and body-fixed flies clearly exhibit distinct head responses (*Figure 3*), but what are the individual contributions of visual and nested mechanosensory feedback underlying these differences? To address this question, we used our mathematical model of gaze stabilization (*Equation 5*) combined with behavioral measurements to predict how visual and mechanosensory feedback individually influence head control.

First, we measured the visual transform $G_{head,V}$, which is the transform between the sensory error $E$ and the head motor response $H$ in body-fixed flies. We then substituted $G_{head,V}$ (see *Figure 5C* for visualization of $G_{head,V}$) into (*Equation 5*) while setting $G_{head,M} = 0$ (indicating no contributions of mechanosensory feedback due to body motion) to predict the effects of body visual feedback on the head motor response (*Figure 4—figure supplement 2A*):

$$H = \underbrace{\frac{G_{head,V}}{1 + G_{head,V}} R}_{\text{head visual feedback}} - \underbrace{\frac{G_{head,V}}{1 + G_{head,V}} B}_{\text{body visual feedback}}.$$

(7)

Using *Equation 7*, we generated a prediction of the closed-loop head transform with body visual feedback. Body visual feedback could partially account for the decrease in magnitude and the increase in phase in body-free flies (*Figure 4C*, *copper*). Notably, head gain at low frequencies shifted from higher gain in body-fixed flies (*Figure 4B*, *purple*) to lower gain when body visual feedback was introduced (*Figure 4C*, *copper*), which more closely matched the body-free head response (*Figure 4B*, *blue*). These visually mediated changes in the head response closely followed the inverse of the body compensation error (*Figure 4A*, *red*), meaning that the more the body reduced sensory error, the smaller the head magnitude became in body-free flies. These results demonstrate that body visual feedback changes the tuning of head responses from broadband (high gain at all frequencies) to high-pass (high gain only at higher frequencies) (*Figure 4C*, *copper* vs. *purple*). Interestingly, body visual feedback could not account for the overall decrease in head gain in body-free flies (*Figure 4A*, *copper* vs *blue*; see *Figure 4—figure supplement 3* for both plots overlaid), suggesting that other sensory modalities must influence head control.

We confirmed our prediction from *Equation 7* by performing a 'replay' experiment, wherein we designed a new visual perturbation for body-fixed flies that had the mean body response of body-free flies subtracted (*Figure 4—figure supplement 2C-E*). In this way, we experimentally reintroduced body visual feedback in body-fixed flies. Due to limitations in spatial resolution of our visual display, we could not properly replay body motion at the two highest frequencies, thus we exclude them. The match between our model prediction (*Figure 4C*, *copper*) and experimental data (*Figure 4C*, *grey*) strongly supports the notion that body visual feedback accounts for the change in tuning across visual motion frequencies—but not the overall decrease in magnitude—of head responses between body-free and body-fixed flies. Furthermore, the close match between model and experiments provides some assurance that the head control system can be modeled with LTI assumptions, thus supporting our LTI-based control theoretic framework.

## Nested mechanosensory feedback damps head movements

If body visual feedback alone cannot fully account for the difference in head motor responses, then it would follow that mechanosensory feedback due to body motion plays a role in shaping head movements. Body visual feedback predicts that body-free flies should have larger head responses at the highest frequency (*Figure 4C*) because body movements increase the sensory error in this range (*Figure 4A*, *red*, compensation error >1), but this was not observed in our experiments, suggesting that there are other sensory modalities at play (*Figure 3C–D*, *Figure 4B*). Prior work showed that flying flies mounted on a motor and rotated about the vertical axis perform compensatory head movements in the opposite direction of the body, even when visual feedback is removed, pointing to the role of haltere-generated mechanosensory feedback (due to body motion) in head control (*Sandeman, 1980*). However, the individual contributions of visual and mechanosensory feedback remain unclear in the control of head movement, particularly due to their nested architecture.

To estimate the contributions of mechanosensory feedback on head control, we compared the transform from $E$ to $H$ in body-free (*Figure 5A*) and body-fixed (*Figure 5B*) flies. In body-fixed flies this transform is purely mediated by visual inputs and equal to $G_{head,V}$, but in body-free flies there is nested mechanosensory feedback that could shape the head response. We defined the $E$ to $H$ transform in body-free flies as $G_{head,V+M}$. We discovered that the gain of $G_{head,V+M}$ was substantially lower, and the phase subtly larger, than $G_{head,V}$, suggesting that nested mechanosensory feedback due to body motion has a transformative influence on head control (*Figure 5C*). By computing the ratio $G_{head,V+M}/G_{head,V}$ we discovered that the overall gain from $E$ to $H$ decreased by a factor of ~0.4 and phase increased by ~20° (*Figure 5D*, *Figure 4—figure supplement 1B*). This ratio describes the effective weighting of nested mechanosensory feedback with respect to visual feedback. These results strongly suggest that the neural signals generated from mechanosensory pathways serve to actively damp head movements. Although this analysis does not isolate the precise sensory mechanism (halteres, antenna, wing proprioceptors, etc.), our findings strongly suggest that the observed change in head control is driven by a mechanosensory modality that measures body motion, thus strongly implicating halteres (*Dickinson, 1999*; *Sherman and Dickinson, 2003*).

Similar to how we could predict the contributions of body visual feedback on the closed-loop head response, we used *Equation 5* to generate a mathematical prediction of the head response with body mechanosensory feedback (*Figure 4B*):

$$H = \underbrace{\frac{G_{head,V}}{1 + G_{head,V}} R}_{\substack{\text{head visual} \\ \text{feedback}}} + \underbrace{\frac{G_{head,M}}{1 + G_{head,V}} B}_{\substack{\text{body mechanosensory} \\ \text{feedback}}} . \qquad (8)$$

However, it was not possible to measure the mechanosensory transform $G_{head,M}$ directly from our experimental data, because visual feedback was always present. Therefore, we derived an expression equivalent to *Equation 8* with $G_{head,V+M}$ in place of $G_{head,M}$ (see Materials and methods):

$$H = \frac{G_{head,V+M}}{1 + G_{head,V+M}} R. \qquad (9)$$

Our prediction of the head response with body mechanosensory feedback due to body motion also partially accounted for the increase in head movement magnitude and decrease in phase in body-fixed flies, but similarly to body visual feedback, could not fully account for the difference (*Figure 4D*, *Figure 4—figure supplement 1A*, *cyan* vs *blue*). Notably, mechanosensory feedback due to body motion led to an overall decrease in head gain across visual motion frequencies in body-free flies (*Figure 4D*, *Figure 4—figure supplement 1A*), whereas visual feedback primarily attenuated low frequency visual motion and changed the tuning of the head response from broadband to high-pass (*Figure 4C*, *Figure 4—figure supplement 1A*). While prior experiments in body-fixed flies mounted on a motor showed that mechanosensory information from the halteres primarily mediates high-frequency steering responses (*Sherman and Dickinson, 2003*), our results strongly suggest that haltere feedback due to body motion has a considerable influence even at lower frequencies. This emergent low-frequency response is likely a property of closed-loop dynamics (due to the body being free to move) that would not be evident in open-loop (body-fixed) conditions. Altogether, out results reveal the precise roles of body-generated visual and mechanosensory feedback in shaping head movement: visual feedback changes the tuning from broad-band to high-pass and mechanosensory feedback reduces the overall magnitude.

## Head damping is present during self-generated but not externally generated body motion

Our findings strongly suggest that the change in the transform from $E$ to $H$ is primarily brought about by a mechanosensory pathway from $B$ to $H$ ($G_{head,M}$). In body-free flies, where $B \neq 0$, this pathway shapes head responses via nested sensory feedback (*Figure 5*). Although it was impossible to measure $G_{head,M}$ directly in body-free or body-fixed flies because head visual feedback was always present, our control framework (*Figure 2A*) allowed us to estimate $G_{head,M}$ and make a prediction of how nested mechanosensory feedback influences head control. We solved for $G_{head,M}$ from *Equation 5*:

$$G_{head,M} = (1 + G_{head,V})\frac{H}{B} - G_{head,V}\frac{R}{B} + G_{head,V}, \qquad (10)$$

which can equivalently be represented as:

$$G_{head,M} = (G_{head,V+M} - G_{head,V})\frac{E}{B}. \qquad (11)$$

Our prediction of $G_{head,M}$ exhibited low gain at low frequencies, but gain swiftly increased with increasing frequency, consistent with previous work that showed that the halteres are most sensitive to high body frequencies/angular velocities in open-loop (*Figure 6A*, *pink*) (*Dickinson, 1999*; *Sherman and Dickinson, 2003*). The shape of $G_{head,M}$ resembled a high-pass filter, suggesting that the halteres may be primarily tuned to angular acceleration in open-loop (*Sandeman, 1980*). This result likely explains why prior studies in body-fixed (open-loop) flies reported that the halteres encode body velocity like a high-pass filter—they were measuring $G_{head,M}$ without self-generated sensory feedback (*Sherman and Dickinson, 2003*). The high gain may seem counter-intuitive—seeing as we argue that mechanosensory feedback actually *decreases* the magnitude of head movements (*Figure 4D*)—however the phase response between –70° and –200° (average –119°) means that the head steering response elicited by mechanosensory feedback destructively interferes (opposite direction) with the visually elicited head steering response, leading to an overall decrease in the magnitude of head movements in body-free flies. An alternate interpretation of these data is that mechanosensory information

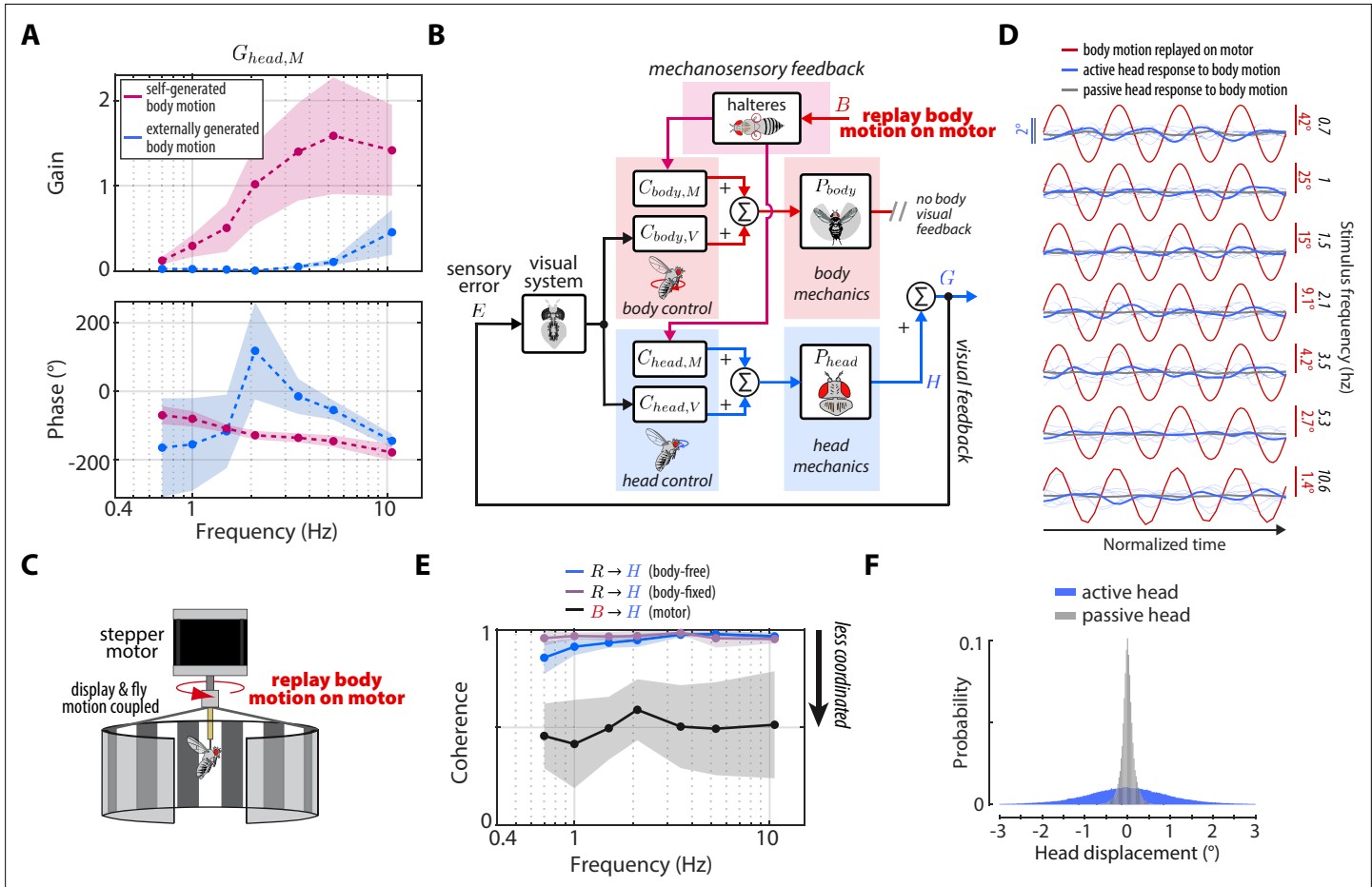

**Figure 6.** Head damping is present during self-generated but not externally generated body motion. (**A**) The predicted transforms from body mechanosensory information to the head response $G_{head,M}$ for self-generated body motion (pink) and externally generated body motion (blue). Shaded regions: ±1 STD. (**B**) The control framework outlining how externally generated body motion influences head response via mechanosensory feedback. Note that head visual feedback is still present even if there is no external visual perturbation since the head is free to move. (**C**) Experimental equivalent to (**B**). Flies were mounted to the shaft of a stepper motor and the body motion measured in body-free flies was replayed on the motor. Note that the visual display was also mounted to the motor shaft, effectively removing body visual feedback, while leaving mechanosensory feedback intact. (**D**) The head response (blue) of flies during the experiment where body motion (red) was replayed on the motor. Thin blue lines show the response of individual flies. Thick grey line shows the mean passive head response of an anesthetized fly to the same replayed body motion. Also see *Figure 6—figure supplement 1*. (**E**) Coherence for the visual transform from $R$ to $H$ in body-free (blue) and body-fixed (violet) flies compared to the mechanosensory transform from $B$ to $H$ measured from the motor experiment. Note that the mechanosensory transform has much lower coherence, indicative of an uncoordinated response. Shaded regions: ±1 STD. (**F**) The distribution of all active head displacements (blue) compared to the distribution of all passive head displacements from the motor experiment (grey). Motor experiments: $n = 6$ flies.

The online version of this article includes the following video and figure supplement(s) for figure 6:

**Figure supplement 1.** Passive head movements.

**Figure 6—video 1.** A fly was mounted on a motor and rotated about the vertical (yaw) axis such that the body angle matched that of a magnetically tethered fly in response to a 1 Hz sine wave visual perturbation (see Figure 3A and *Figure 3—video 1*).
https://elifesciences.org/articles/80880/figures#fig6video1

**Figure 6—video 2.** Same as *Figure 6—video 1* but for a 2.1 Hz perturbation.
https://elifesciences.org/articles/80880/figures#fig6video2

**Figure 6—video 3.** Same as *Figure 6—video 1* but for a 5.3 Hz perturbation.
https://elifesciences.org/articles/80880/figures#fig6video3

**Figure 6—video 4.** Same as *Figure 6—video 1* but for an anesthetized fly.
https://elifesciences.org/articles/80880/figures#fig6video4

**Figure 6—video 5.** Same as *Figure 6—video 2* but for an anesthetized fly.
https://elifesciences.org/articles/80880/figures#fig6video5

*Figure 6 continued on next page*

*Figure 6 continued*

https://elifesciences.org/articles/80880/figures#fig6video5

**Figure 6—video 6.** Same as *Figure 6—video 3* but for an anesthetized fly.

https://elifesciences.org/articles/80880/figures#fig6video6

is subtracted from, rather than added to (as in *Figure 3A*), visual information in the nervous system (i.e., negative feedback).

An interesting idea to consider is that the damping due to mechanosensory feedback we uncovered is only present during self-generated (i.e., nested) rather than externally generated body motion (e.g. from a gust of wind). The yaw flight axis is inherently stable—as opposed to pitch—so flies may only require mechanosensory feedback during self-generated yaw turns, where flies need to damp out their own motion (*Taha et al., 2020*; *Faruque and Sean Humbert, 2010a*; *Faruque and Sean Humbert, 2010b*). Although this idea mainly applies to the control of body movements, the head may be controlled similarly. To this end, we designed an experiment where we imposed externally generated body motion with no body visual feedback to uncover $G_{head,M}$ for externally generated, rather than self-generated body motion (*Figure 6B*). We mounted rigidly tethered flies to the shaft of a stepper motor and replayed the recorded body motion of body-free flies $B$, while measuring their head responses $H$ (*Figure 6C*). Crucially, the visual display was also fixed to the motor shaft, so as to remove any visual feedback generated from body motion.

Intriguingly, the head response to externally generated body motion was small and generally uncoordinated with body motion (*Figure 6D*, *Figure 3—video 4–Figure 6—video 2*). We computed the coherence—a measure of linear correlation in frequency domain where values near one indicate high correlation and values near zero indicate low correlation—between the externally generated body motion and the head response and found that the head operated with a coherence of ~0.5. Compared to the head responses driven by visual motion—which operated with coherence near 1—our results demonstrate that flies do not have a robust head response to externally generated body motion about the yaw axis (*Figure 6C*), corroborating previous work that measured wing movements (*Sherman and Dickinson, 2003*). We computed $G_{head,M}$ from these experiments using our control framework:

$$G_{head,M} = (1 - G_{head,V})\frac{H}{B}, \tag{12}$$

and compared the response to $G_{head,M}$ computed for self-generated body motion (*Equation 10–Equation 11*). We discovered that $G_{head,M}$ for externally generated body motion displayed gains nearly an order of magnitude smaller than $G_{head,M}$ for self-generated body motion (*Figure 6A*). The phase estimates were highly variable due to the low-coherence response (*Figure 6A*). These findings strongly suggest that mechanosensory information is integrated in a nonlinear fashion, that is dependent on the type of body motion: externally- vs self-generated. The precise mechanism that underlies the gating is unclear, although it is likely that self-generated turns evoke mechanosensory-dependent activity of muscles within the neck motor system (*Huston and Krapp, 2009*).

## Mechanical properties of the neck joint prevent passively generated head motion

To ensure that the head responses we measured in flies mounted on the motor and in the magnetic tether (where the body also moves the same way) were elicited by sensory feedback—not generated mechanically from body motion—we repeated the same experiment illustrated in *Figure 6C*, but for anesthetized flies. This approach allowed us to isolate any passively generated head movements due to body motion that were not under active neural control. We found that passively generated head movements were much smaller than head movements of actively flying flies (*Figure 6D*, *grey* vs *blue*, *Figure 6—videos 4–6*). The head rarely moved more than 0.5° in anesthetized flies, compared to 2° in active flies, demonstrating that sensory feedback is the primary driver of head movements (*Figure 6F*). This was consistent for a sum-of-sines replay experiment (*Figure 6—figure supplement 1*). Interestingly, the passive mechanics of the neck joint (stiffness, damping, etc.) effectively decoupled the head from the body, which could simplify the neural control of head movements because flies would not have to account for passive head motion (*Cellini et al., 2021*). Neck passive mechanics

could also help keep the head stable during rapid turns or large external perturbations such as turbulent gusts of wind.

## Head saccades are actively damped by mechanosensory feedback

Our results thus far strongly implicate mechanosensory feedback due to body motion in damping smooth head responses during self-generated, but not externally generated turns. However, in addition to the smooth head and body movements during gaze stabilization, flies also perform rapid, saccadic turns of the head and body (*Cellini and Mongeau, 2020b*; *Cellini et al., 2021*; *Mongeau and Frye, 2017*; *Collett and Land, 1975*; *Bender and Dickinson, 2006b*; *Muijres et al., 2015*). Prior studies suggest that, once executed, body saccades are visually open-loop, as body saccade duration is on the same order as visuomotor delays and altering visual feedback during body saccades does not change their dynamics (*Bender and Dickinson, 2006a*; *Mongeau and Frye, 2017*). However, mechanosensory feedback is thought to play a role in eliciting the wing-braking response to terminate body saccades (*Cellini and Mongeau, 2020b*; *Bender and Dickinson, 2006a*). Head saccades are thought to be similarly visually open-loop (*Kim et al., 2017*; *Cellini et al., 2021*). However, as prior work has shown that visual and mechanosensory inputs converge at the neck motor center (*Milde et al., 1987*; *Strausfeld and Seyan, 1985*), we hypothesized that mechanosensory feedback due to body motion also influences head saccade dynamics. Specifically, due to the damping effects of mechanosensory feedback we uncovered during self-generated body motion, we predicted that head saccades in body-free flies should be of smaller magnitude than in body-fixed flies.

To test this hypothesis, we culled head saccades from body-free and body-fixed flies presented with a static visual stimulus using a previously described method (*Figure 7A–B*, *Figure 7—video 1*, *Cellini et al., 2021*; *Mongeau and Frye, 2017*; *Salem et al., 2020*). Consistent with our prediction, head saccades in body-free flies displayed smaller amplitude and peak velocity than head saccades in body-fixed flies, suggesting that mechanosensory feedback damps head saccades (*Figure 7C–D*), as it does for whole-body saccades (*Bender and Dickinson, 2006a*). Interestingly, head saccades in body-free flies were also immediately followed by a head movement that returned the head to the neutral position (*Figure 7C*). However, this return head movement was absent, or much slower, in body-fixed flies, suggesting that mechanosensory feedback plays an important role in terminating, or braking, head saccades (*Figure 7C*). By fitting a decaying exponential (total damping time constant) to the head trajectory immediately after the head saccade responses, we discovered that body-fixed flies took ~8 times longer to return to baseline than body-free flies (Wilcoxon rank sum, p < 0.001) (*Figure 7D*). Interestingly, during the return head movement in body-free flies, the body was still in motion (*Figure 7C*), suggesting that body-generated feedback, or lack thereof, is the mechanism driving this difference in behavior. Because visual sensory feedback has little effect on saccade dynamics (*Bender and Dickinson, 2006a*), this damping of head saccades is likely driven by nested mechanosensory feedback—although some degree of passive (mechanical) damping is likely present as well (*Cellini et al., 2021*). We found that head saccades performed in dark visual conditions followed similar trajectories, supporting the notion that mechanosensory, not visual, feedback mediates head saccade damping (*Figure 7—figure supplement 1*). Intriguingly, the decrease in damping in body-fixed flies could also explain why wing saccades last upwards of 500ms in body-fixed flies, while body saccades in free flight typically last only 50–100ms (*Cellini and Mongeau, 2020b*). While prior work has demonstrated that local haltere and wing muscle proprioceptive feedback feedback influence visuomotor gain (*Kathman and Fox, 2019*; *Mureli and Fox, 2015*; *Mureli et al., 2017*; *Bartussek and Lehmann, 2016*; *Lehmann and Bartussek, 2017*), it is unlikely that this mechanism could explain the attenuated saccade dynamics, due to saccades being visually open-loop (*Bender and Dickinson, 2006a*). Overall, our findings strongly suggest that nested mechanosensory feedback has a significant influence on the control of both smooth head movements and head saccades in flies.

## Discussion

We developed a mathematical model of gaze stabilization that accounted for the role of visual feedback and nested mechanosensory feedback in mediating head responses in flies (*Figure 2*). Our model predicted differences in head responses between body-free and body-fixed flies based on changes in sensory feedback, which we confirmed with experimental data (*Figure 3*). We revealed

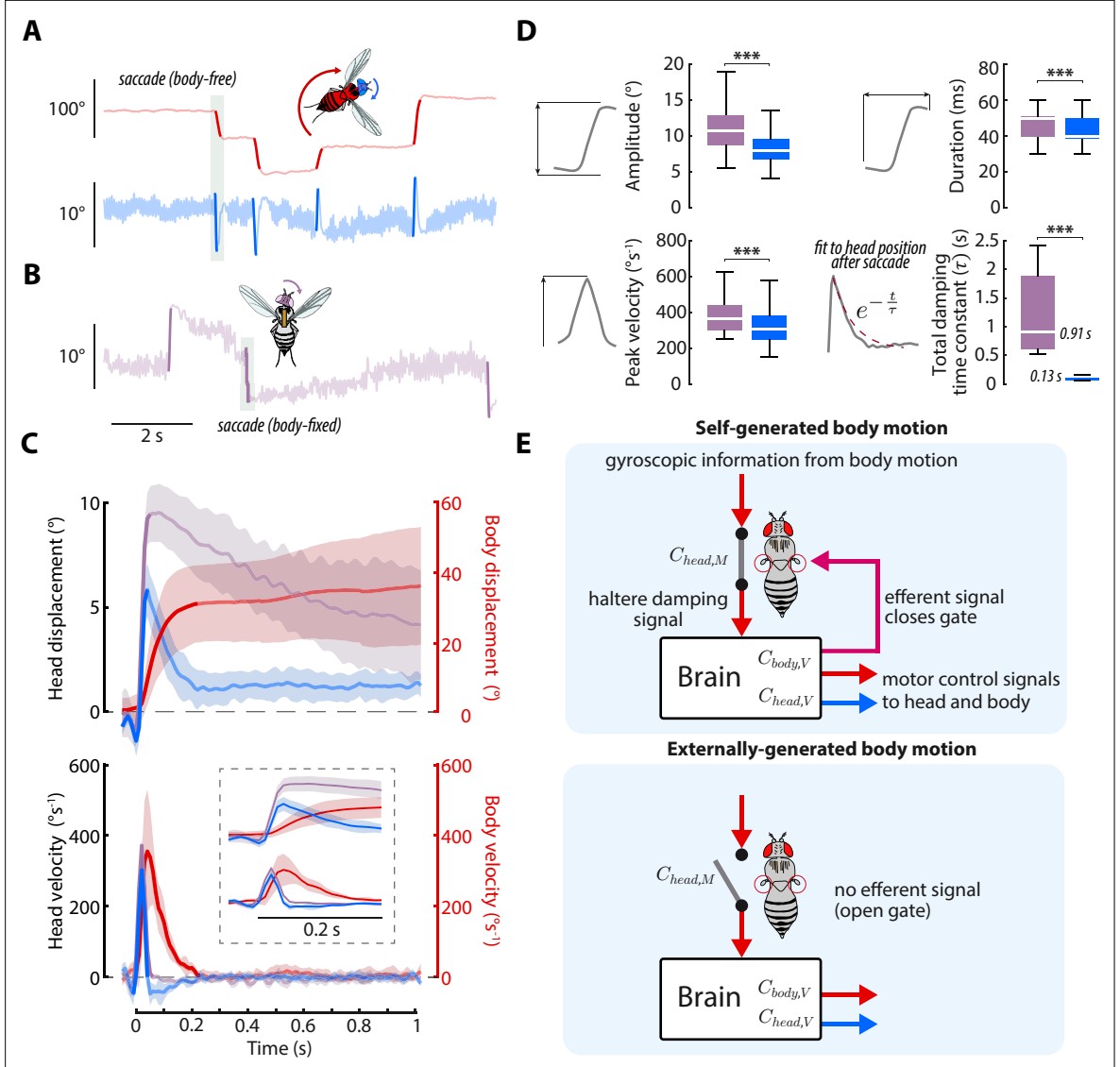

**Figure 7.** Head saccades are actively damped by mechanosensory feedback. (**A**) Example body (red) and head (blue) trajectories for a body-free fly in the magnetic tether presented with a static visual stimulus. Rapid flight turns called saccades are highlighted. Note that head saccades are followed by a head movement that returns the head to the center position. Also see ***Figure 7—video 1***. (**B**) Same as A) but for a body-fixed fly. Head movements are shown in purple. Note that head saccades are not followed by a return head movement. (**C**) Left y-axis: averaged head saccade displacement (top) and velocity (bottom) for body-free and body-fixed flies. Right y-axis: averaged body saccade displacement (top) and velocity (bottom). Note that saccades typically last less than 200ms (bold portion of head and body trajectories indicate saccades), but an extra second of data is shown to illustrate the difference between the body-free and body-fixed head movements after a saccade. Inset shows the first 200ms of head and body trajectories. Shaded regions: ±1 STD. (**D**) Distributions of head saccade amplitude, peak velocity, duration, and damping time constant. The damping time constant $\tau$ was computed by fitting a decaying exponential to the head response directly after a saccade. ***Wilcoxon rank sum and t-test, p < 0.001. Body-free: $n = 8$ flies, $N = 566$ saccades, Body-fixed: $n = 8$ flies, $N = 346$ saccades. (**E**) Proposed neural architecture for haltere-related damping of head movements for self-generated vs. externally-generated body motion. When body motion is self-generated, head and body motor commands are sent in parallel with an efferent signal, effectively closing a gate that allows mechanosensory feedback due to body motion to damp head movements. When body motion is externally generated, this gate is open and body motion has little effect on head movements (***Figure 6A and (D)***).

The online version of this article includes the following video and figure supplement(s) for figure 7:

**Figure supplement 1.** Head saccades in darkness.

**Figure 7—video 1.** A body-free fly (magnetic tether) presented with a static visual background performing simultaneous body and head saccades.

https://elifesciences.org/articles/80880/figures#fig7video1

that visual feedback influenced the frequency tuning of head movements, whereas nested mechanosensory feedback due to body motion reduced the overall magnitude of head responses during smooth movements and saccades via active damping (*Figures 4–7*). By comparing head responses during self-generated and externally generated body motion, we uncovered a nonlinear gating of body-generated mechanosensory feedback on head movements influenced by self-motion. Overall, our findings unravel multisensory integration within nested sensory feedback loops in insect flight. We provide a framework amenable to study nested biological feedback loops across phyla.

## Change in head movements between body-free and body-fixed flies is an emergent property of reflexive feedback

We discovered that body-fixed flies exhibited exaggerated head movements compared to body-free flies, which mirrors their exaggerated wing movements (*Fry et al., 2005*). At face value, it might appear that body-fixed flies are adapting to the lack of stabilizing body movements by moving their head with larger magnitude. However, such a mechanism implies that flies must learn and change their neural controller ($C_{head,V}$ or $C_{head,M}$ in *Figure 2A–B*), to compensate for body fixation. Instead, our results support the notion that visual feedback underlies these changes and enables flies to partially compensate for body fixation. In other words, the larger head movements observed in body-fixed flies are an emergent property of reflexive feedback. In essence, the increase in sensory error due to body fixation (*Figure 3*) elicits a larger head motor response immediately, without requiring flies to learn a new neural controller. In this way, flies have some degree of built-in redundancy in their gaze stabilization system. An emergent property of this type of system is robustness to changes in the dynamics of one of the 'motors' (head or body). For example, wing damage is a common injury experienced by insects which could impair their ability to stabilize gaze via body movements during flight (*Rajabi et al., 2020*). The change in visual feedback following wing damage could enable insects to rapidly compensate with their head, rather than learn how to stabilize gaze with a damaged wing (*Muijres et al., 2017*). Indeed, we show that the head's performance improves at low frequencies when the body is fixed (*Figure 5B–C*). This idea is further supported by behavioral evidence in primates, where eye movements in monkeys increase in magnitude to compensate for sudden head fixation with no detectable change in gaze (head +eye) velocity (*Bizzi, 1981*; *Lanman et al., 1978*). Tuned sensory feedback can be preferable to learning because sensory feedback acts on the order of neural latency and does not require animals to learn new strategies. However, sensory feedback and learning are not mutually exclusive and flies likely exhibit both (*Wolf et al., 1992*).

## Nested mechanosensory feedback actively damps head movements

Mechanosensory feedback plays an important role in flight stability. Flies with bilateral haltere ablations or immobilized halteres cannot achieve stable flight (even in magnetically tethered assays), directly implicating mechanosensory feedback from the halteres in flight stabilization (*Ristroph et al., 2013*). For the body, it appears that the low (~5ms) sensory delays associated with the halteres act synergistically with the slower (~30ms) visual system (*Sherman and Dickinson, 2003*; *Nakahira et al., 2021*), thereby permitting larger visual gains (*Elzinga et al., 2012*). In hawk moths, mechanosensory feedback from the antennae is nested within visual feedback during flower tracking, which is critical for high-frequency performance (*Dahake et al., 2018*). This suggests that the structure of visuo-mechanosensory integration may be a preserved feature of insect flight. Visuo-mechanosensory integration also likely explains why wing responses are exaggerated in body-fixed flight, because mechanosensory feedback is not present to actively damp out steering responses (*Elzinga et al., 2012*; *Taylor et al., 2008*). However, the role of mechanosensory feedback and stability is less clear when considering the control of head movements. Indeed, the biomechanics of the head-neck system are inherently stable (*Cellini et al., 2022*; *Cellini et al., 2021*), so what is the role of mechanosensory feedback in the head control system?

About the roll and pitch axes, head movements serve to maintain level gaze by offsetting variations in body roll and pitch (*Hardcastle and Krapp, 2016*; *Hengstenberg, 1984*). However, we show that this is largely not the case for about the yaw axis during externally generated body movements (*Figure 6*), suggesting that there is another mechanism at play. We discovered that mechanosensory feedback actively damped head movements in body-free flies (*Figure 5*, *Figure 6*), similar to the proposed active damping of wing movements (*Elzinga et al., 2012*). The active damping of head

movements decreased head excursions, and occurrences of head saturation were reduced to near zero (*Figure 3C and D*). An interesting possibility is that mechanosensory feedback from body movements may act as a centering reflex to keep the head aligned relative to the thorax and thus prevent the head from reaching large angular excursions. Indeed, gaze stabilization quickly degrades as the head reaches its anatomical limits (*Cellini et al., 2021*). Another potential explanation is that the effects of mechanosensory feedback on head control are simply a result of coupled neural pathways between body control and head control. The descending sensorimotor pathways associated with the head and body have some overlap, suggesting that similar information—such as damping commands from the halteres—could be shared between the head and body (*Namiki et al., 2018*). Revealing the precise role of mechanosensory feedback will require analyses of the neural pathways associated with head and body control.

## Nested proprioception across phyla

Our work in flies shows that sensing body motion via mechanosensory feedback has a transformative influence on a task that is driven by visual inputs (optomotor response). But similar proprioceptive mechanisms exist across phyla, and nested proprioception is likely a prevalent feature of animal locomotion. Many locomotor tasks in which an animal's whole body moves in response to some external stimuli—such as flower tracking in moths (*Sponberg et al., 2015*; *Roth et al., 2016*), refuge tracking in fish (*Roth et al., 2011*; *Uyanik et al., 2020*), and wall following in cockroaches (*Cowan et al., 2006*; *Mongeau et al., 2015*)—likely involve proprioceptive feedback. Our framework could be applied to tease apart the role of proprioceptive mechanisms—such as the role of antennae, the vestibular system, or other mechanosensors—in task-level control. For instance, flies appear to use their antenna to damp out their visually guided groundspeed controller in a nested fashion (*Fuller et al., 2014a*). A comparable experiment in mice or fish, where vestibular feedback from the the inner-ear is abolished (via chemical labyrinthectomy) could provide insights into how proprioception shapes locomotion in vertebrates (*Ito et al., 2019*). Altogether, our framework is generalizable for teasing out the role of nested proprioception in a range of animal behaviors.

## Distinguishing between self-generated and externally generated body motion

Mechanosensory feedback due to body motion had little influence on head movements when body motion was externally generated as opposed to self-generated (*Figure 6*). For a LTI system, one would expect the same sensory inputs to lead to the same outputs. However, flies did not follow this expectation, suggesting that motor-related signals or visual feedback gate (non-linearly) mechanosensory feedback. We propose a model in which self-generated head/wing steering commands are sent in parallel with a signal that opens a gate to allow mechanosensory information to flow to the neck motor center (*Figure 7E*). One possible mechanism at the neural level is that flies actively modulate gyroscopic sensing via haltere steering muscles.

Recent work confirmed that the haltere muscles are actively modulated by visual inputs during flight (*Dickerson, 2020*). The "control-loop" hypothesis—originally proposed by *Chan et al., 1998*—suggests that visual inputs modulate haltere muscle activity, which then regulate mechanosensory feedback by recruiting haltere campaniform sensilla (*Chan et al., 1998*; *Dickerson et al., 2019*). One possibility is that visual inputs could modulate haltere muscle activity and increase the magnitude of gyroscopic inputs, thus leading to damped head dynamics. Due to the body-fixed assays required for electrophysiology, it has not been impossible to determine whether gyroscopic inputs are modulated by visual inputs, but our results suggest that this could be the case. Altogether, our findings provide an extension of the control-loop hypothesis for the more specific case of gyroscopic sensing, that distinguishes between motor context (self-generated vs. externally generated body motion).

We cannot discount other mechanisms—such as haltere afferents gating subpopulations of neck motor neurons' responses to visual stimuli—as the integration of visual and mechanosensory information is often nonlinear in insect flight (*Sherman and Dickinson, 2004*; *Huston and Krapp, 2009*; *Haag et al., 2010*; *Kathman and Fox, 2019*). Alternatively, gating may be modulated by an efference copy during self-generated turning maneuvers. These two hypotheses could not be teased apart here because flies mounted to a motor almost immediately stop flight if visual inputs conflict with prescribed motor rotation, that is, it was necessary to mount the visual display to the motor shaft for

flies to sustain flight, thereby eliminating retinal slip due to externally generation motion, as in prior work (*Sherman and Dickinson, 2003*).

## Neurophysiological evidence for gating of visual and mechanosensory information

The nonlinear gating we described here corroborates neurophysiological data on the influence of haltere *tonic* inputs on neck motor neurons. Recordings from a subpopulation of neck motor neurons demonstrated that information from the eyes and halteres is combined nonlinearly (*Huston and Krapp, 2009*). Specifically, some neck motor neurons do not generate action potentials in response to visual motion alone, but will generate action potentials when the halteres are beating simultaneously and providing tonic inputs. Furthermore, the ventral cervical nerve motoneuron (VCNM) cell—which mediates head control—receives input from visual, haltere, and antennal sensory neurons (*Haag et al., 2010*). Visual motion alone generates subthreshold activity, but when combined with mechanosensory inputs (antennae or halteres), causes the VCNM to spike. Notably, VCNM integrates a central input reflecting the behavioral state of the fly (flight and non-flight). While the influence of haltere tonic activity on downstream circuits has been characterized, at present it is unclear how *gyroscopic* inputs from the halteres influence neck motor neurons, primarily due to technical limitations of using fixed neurophysiological preparations. An interesting possibility is that some neck motor neurons, in the presence of gyroscopic feedback, could actively brake head movements thus providing a mechanism for active damping.

## Materials and methods
### Animal preparation

We prepared flies according to a previously described protocol (*Cellini et al., 2022*; *Cellini and Mongeau, 2020a*). Briefly, we cold-anesthetized 3- to 5-day-old females flies (wild-type *Drosophila melanogaster*) by cooling them on a Peltier stage maintained at ~4° C.Following cold anesthesia, we fixed stainless steel minutien pins (100 μm diameter, Fine Science Tools, Foster City, CA) to the thorax of each fly using UV-activated glue (XUVG-1, Newall). We fixed the pin at angle of ~30°, consistent with the body's angle of attack in freely flying flies. We allowed ~1 hr for recovery. For the body-free condition, we suspended each fly between two magnets, allowing free rotation along the yaw (vertical) axis (*Figure 2C*). The pin was fit into a sapphire bearing which has a coefficient of friction of ~0.1 (Vee jewel bearing, Bird Precision), which flies can readily overcome (*Cellini et al., 2022*). The inertia of the pin was less than 1% of the fly's inertia. Further, using an electromagnetic simulation we previously showed that frictional forces due to the pin-bearing interface are about two orders of magnitude smaller than forces generated in flight (*Cellini et al., 2022*). Thus flies can readily overcome this friction, as previously shown (*Mongeau and Frye, 2017*). For rigidly tethered (body-fixed) flies, all preparations were the same except we fixed flies to tungsten pins (A-M Systems) which were rigidly held in place (*Figure 2D*). This is in contrast to previous work that instead rigidly tethered flies with a 90° angle with respect to the pin and angled the pin itself to 30°, although this difference has little effect on the head response (*Cellini and Mongeau, 2020a*; *Cellini et al., 2021*).

### Flight simulator

The virtual reality flight simulator illustrated in *Figure 2C–D* has been described elsewhere (*Mongeau and Frye, 2017*; *Reiser and Dickinson, 2008*). The display consists of an array of $96 \times 16$ light emitting diodes (LEDs, each subtending 3.75° on the eye) that wrap around the fly, subtending 360° horizontally and 60° vertically. We recorded the voltage signal output from our flight arena's visual display with a data acquisition system (Measurement Computing, USB-1208FS-PLUS), which measures the displacement of our prescribed visual perturbation. We used this signal as the input to our model to ensure we accurately quantified what flies were actually seeing during experiments. We placed flies in the center of the arena and provided illumination from below with an array of twelve 940 nm LEDs and two 940 nm LEDs above. Body-free and body-fixed flies were examined using the same flight simulator. We recorded video at 100 $frames \cdot s^{-1}$ with an infrared-sensitive camera placed directly below the fly (Basler acA640–750 μm). We used our custom computer vision software suite, CrazyFly (https://github.com/boc5244/CrazyFly, *Cellini, 2021*) to analyze body and head kinematics. The tracking

algorithms have been described elsewhere (*Cellini et al., 2022*). We measured the body angular position with respect to a global (flight arena) coordinate frame and head angular position relative to the body in each frame in our recorded videos.

## Visual perturbations

We primarily employed seven previously constructed single-sines visual perturbations $R(t)$ at frequencies $f = [0.7, 1, 1.5, 2.1, 3.5, 5.3, 10.6]$ Hz, designed to elicit robust head and body responses across a broad frequency range (*Cellini et al., 2022*). The amplitude $A$ at each frequency was chosen such that the velocity was normalized to $\dot{R}_{norm} = 250°s^{-1}$ (mean speed of $159°s^{-1}$). The perturbations can be represented as:

$$R(t) = \frac{\dot{R}_{norm}}{2\pi f_i} \sin(2\pi f_i t). \tag{13}$$

As in prior work, we also employed three previously constructed sum-of-sines visual perturbations (*Cellini et al., 2022*). Briefly, each visual perturbation consisted of a 20-s sum-of-sines signal with nine logarithmically spaced frequencies components $f_i$ in increments of 0.05 Hz, where no frequency was a prime harmonic of another. The amplitude of each frequency component was chosen such that the velocity of each component was normalized to thee values of $\dot{R}_{norm} = [42, 70, 95]°s^{-1}$ and the phase $\phi_i$ was randomized. The perturbations can be represented as:

$$R(t) = \sum_{i=1}^{9} \frac{\dot{R}_{norm}}{2\pi f_i} \sin(2\pi f_i t + \phi_i). \tag{14}$$

These single-sine and sum-of-sines visual perturbations cover the frequency range of natural scene dynamics that a fly would normally experience in free flight (*Kern et al., 2005*). All visual perturbations were displayed on our flight simulator as a grating with 30° spatial wavelength. This ensured that our perturbations had a mean temporal frequency of ~5 Hz, near the optimum of the motion vision pathway in *Drosophila* (*Figure 2C–D*, *Jung et al., 2011*).

## Non-parametric system identification

Using a previously described method (*Cellini et al., 2022*), we applied frequency-domain system identification to determine non-parametric frequency-response functions from behavioral data. For a given input (ex: visual perturbation $R$ or sensory error $E$) and output (ex: head $H$ or body $B$) signal, we aimed to determine the relative magnitude (gain) and timing (phase) in frequency domain. We first detrended and low-pass filtered (cutoff 40 Hz) each signal in time-domain to remove low-frequency drift and high-frequency noise. We then transformed the input and output signals into frequency domain using a Chirp-Z Transform (*Remple and Tischler, 2006*; *Windsor et al., 2014*) at frequency points between 0–50 Hz in increments of 0.05 Hz. We divided the resulting complex response of the output signal by the complex response of the input signal, resulting in the frequency-response function $X(s)$ describing the transformation between input and output. We made no explicit assumption of linearity in $X(s)$, as flies' visuomotor responses tend to be marginally nonlinear (*Cellini et al., 2022*; *Cellini and Mongeau, 2020a*). However, the high coherence or our transforms (*Figure 4—figure supplement 3*), the consistent responses between single-sine and sum-of-sines perturbations (*Figure 4—figure supplement 3* vs *Figure 4—figure supplement 1*), and the close match between our replay experiment and theoretical prediction (*Figure 4C*, *Figure 4—figure supplement 3*) suggests that linear techniques can still be used with a high degree of confidence.

We extracted the gain and phase by taking the magnitude $|X(s)|$ and angle $\angle X(s)$ of the complex response, respectively. We calculated the compensation error for each closed-loop frequency response function by computing the distance between the $X(s)$ and the perfect compensation condition $1 + 0j$ (gain = 1, phase = 0°) on the complex plane (*Cellini et al., 2022*; *Sponberg et al., 2015*). Compensation error can be expressed as:

$$\epsilon = |(1 + 0j) - X(s)|. \tag{15}$$

We also calculated the coherence of each closed-loop transform using the MATLAB routine *mscohere* to ensure that head and body movements were sufficiently related to the visual perturbations (*Figure 4—figure supplement 3*, *Figure 4—figure supplement 1*).

Wherever there was saturation in the head response for body-fixed flies (as in *Figure 3C*, top), we applied saturation-corrected least-squares-spectral-analysis (LSSA) (*Figure 3—figure supplement 2*). Briefly, we removed the saturated portion of the data (where velocity was near zero) and fit a sine wave to the remaining un-saturated data (*Figure 3—figure supplement 2*). Then we corrected the gain of any transforms affected by the saturation ($R \to H$ and $E \to H$ for body-fixed flies) (*Figure 3—figure supplement 2A*). To confirm that LSSA itself was not changing our results, we compared all transforms for the Chirp-Z transform and LSSA (without saturation correction) methods. Both methods yielded virtually identical results (*Figure 3—figure supplement 2B*). Only the lowest two frequencies showed any difference in gain after the saturation correction routine and phase was unaffected across all frequencies and all methods (*Figure 3—figure supplement 2B*).

## Uncertainty propagation in frequency response functions

Experimentally measured frequency-response functions—such as the transforms between $R$ and $H$ in body-free and body-fixed flies shown in *Figure 4* and the $E$ to $H$ transforms in *Figure 5* —were measured for each fly and the mean and standard deviation of the gain, phase, and compensation error were calculated across flies (using circular statistics for phase *Berens, 2009*). However, we were not able to apply the same statistical framework to estimate confidence intervals when computing mathematical predictions, such as in *Equation 7*, because our derived equations combined data sets with different groups of flies. For example, $G_{head,V}$ was measured in body-fixed flies, but $B$ in body-free flies. Therefore, we estimated confidence intervals using a propagation gaze of uncertainly analysis as described in prior work (*Roth et al., 2016*).

## Stepper motor experiments

Flies were rigidly tethered to the shaft of a stepper motor (Nema 17) (*Figure 6C*). The stepper motor was controlled with a motor driver (TB6600) with a resolution of 0.225° per step, thus providing smooth motion of the body. We controlled the motor by sending step and direction signals to the driver from a DAQ. We printed a black and white grating with 30° spatial wavelength (matching the grating displayed on our flight simulator) on standard A4 paper. We fixed the grating in a circular pattern to the motor shaft using a custom 3D printed part (*Figure 6C*). This ensured that any rotations of the motor—and thus the fly's body—did not induce any visual feedback. We replayed the mean body motion measured from actively flying flies in the magnetic tether (*Figure 3A*, *red*) on the motor and measured the corresponding head response for actively flying flies (*Figure 6D*, *blue*) and anesthetized flies (*Figure 6D*, *grey*). We used triethylamine (commercially available as FlyNap, Carolina Biological Supply) to anesthetize flies. Also see *Figure 6—figure supplement 1* for the passive head response to a sum-of-sines perturbation.

## Control framework and derivation of closed-loop head responses

We synthesized our control framework based on previous work on the control of head and body movements in flies, where head and body velocity are the state variables (*Cellini et al., 2022*). However, we made all computations based on head and body displacements, as taking the derivative (i.e. computing the velocity) of complex signals does not change the mathematical relationship between such signals (i.e. gain and phase stay the same). Furthermore, numerical differentiation typically amplifies noise (*van Breugel et al., 2020*). When computing complex valued transforms (e.g., $G_{head,V}$), we calculated the gain and phase for each frequency of the visual perturbation and constructed a non-parametric curve that consisted of the collection of these gains and phase values. All algebra done with these complex valued transforms was done by converting the gain $|X(s)|$ and phase $\angle X(s)$ into a single complex number $X(s) = |X(s)|e^{j\angle X(s)}$ and substituting into the expressions we derive below.

To derive the expressions for the head response under different sensory feedback conditions, we started by considering the body-free case where sources of feedback are present. Head motion $H$ can be written as the sum of sensory error $E$ multiplied by the visual transform $G_{head,V}$ and body motion $B$ multiplied by the mechanosensory transform $G_{head,M}$ based on *Figure 2A*:

$$H = G_{head,V}E + G_{head,M}B, \tag{16}$$

where $E$ is equivalent to the visual perturbation $R$ subtracted by the fly's gaze (sum of $H$ and $B$):

$$E = R - H - B. \tag{17}$$

This framework omits the role of neck proprioceptive feedback, as the neck sensory system is intact in both body-free and body-fixed flies. Substituting *Equation 17* into *Equation 16* yields:

$$H = G_{head,V}(R - H - B) + G_{head,M}B = G_{head,V}R - G_{head,V}B + G_{head,M}B - G_{head,V}H, \tag{18}$$

where we can solve for $H$ to obtain *Equation 5*. Normalizing by $R$ yields the closed-loop transform from $R$ to $H$ that we show in *Figure 4*:

$$\frac{H}{R} = \underbrace{\frac{G_{head,V}}{1 + G_{head,V}}}_{\substack{\text{head visual}\\\text{feedback}}} - \underbrace{\frac{G_{head,V}}{1 + G_{head,V}}\frac{B}{R}}_{\substack{\text{body visual}\\\text{feedback}}} + \underbrace{\frac{G_{head,M}}{1 + G_{head,V}}\frac{B}{R}}_{\substack{\text{body mechanosensory}\\\text{feedback}}}. \tag{19}$$

Because the body also has its own associated visual and mechanosensory transforms $G_{body,V}$ and $G_{body,M}$ (*Figure 2A*), we could remove the $B$ term from *Equation 5* and *Equation 19* by substituting $G_{body,V}$ and $G_{body,M}$. However, the resulting expression is lengthy and does not provide intuitive insights into the different sources of feedback as in *Equation 5* and *Equation 19* , therefore we chose not to include it.

To obtain the body-fixed closed-loop transform corresponding to *Equation 6*, we set $B = 0$ in *Equation 19*:

$$\frac{H}{R} = \underbrace{\frac{G_{head,V}}{1 + G_{head,V}}}_{\substack{\text{head visual}\\\text{feedback}}}. \tag{20}$$

To obtain the closed-loop transform without body mechanosensory feedback (body and head visual feedback only) we set $G_{head,M} = 0$ in *Equation 19*:

$$\frac{H}{R} = \underbrace{\frac{G_{head,V}}{1 + G_{head,V}}}_{\substack{\text{head visual}\\\text{feedback}}} + \underbrace{\frac{G_{head,V}}{1 + G_{head,V}}\frac{B}{R}}_{\substack{\text{body visual}}}. \tag{21}$$

To obtain the closed-loop transform without body visual feedback (body mechanosensory feedback and head visual feedback only) we modified *Equation 17* such that $E = R - H$ and re-derived *Equation 19*, effectively removing the second term:

$$\frac{H}{R} = \underbrace{\frac{G_{head,V}}{1 + G_{head,V}}}_{\substack{\text{head visual}\\\text{feedback}}} - \underbrace{\frac{G_{head,M}}{1 + G_{head,V}}\frac{B}{R}}_{\substack{\text{body mechanosensory}\\\text{feedback}}}. \tag{22}$$

However, there is an interesting trick where we use the transform from $E$ to $H$ ($G_{head,V+M}$, see *Figure 5A*) to simplify this expression and remove the $G_{head,M}$ term. $H$ can be written equivalently to *Equation 16* as:

$$H = G_{head,V+M}E. \tag{23}$$

By substituting *Equation 17* (with $B = 0$) into *Equation 23* and solving for $H$, we obtain *Equation 9* which we can normalize by $R$ to obtain:

$$\frac{H}{R} = \frac{G_{head,V+M}}{1 + G_{head,V+M}}. \tag{24}$$

When making predictions using *Equation 7*, *Equation 9*, and *Equation 12*, we used data-driven methods rather than fitting closed-form transfer function models to the data. Thus, our transforms in *Figures 4–6* do not explicitly assume a model order.

## Acknowledgements

We thank Mark Frye and Martha Rimniceanu for valuable comments. This material is based upon work supported by the Air Force Office of Scientific Research (FA9550-20-1-0084) and an Alfred P Sloan Research Fellowship to JMM.

## Additional information

### Funding

| Funder | Grant reference number | Author |
|---|---|---|
| Air Force Office of Scientific Research | FA9550-20-1-0084 | Jean-Michel Mongeau |
| Alfred P. Sloan Foundation | FG-2021-16388 | Jean-Michel Mongeau |

The funders had no role in study design, data collection and interpretation, or the decision to submit the work for publication.

### Author contributions

Benjamin Cellini, Conceptualization, Data curation, Formal analysis, Investigation, Methodology, Writing - original draft, Writing - review and editing; Jean-Michel Mongeau, Conceptualization, Supervision, Funding acquisition, Methodology, Writing - original draft, Writing - review and editing

### Author ORCIDs

Benjamin Cellini http://orcid.org/0000-0002-0609-7662
Jean-Michel Mongeau http://orcid.org/0000-0002-3292-6911

### Decision letter and Author response

Decision letter https://doi.org/10.7554/eLife.80880.sa1
Author response https://doi.org/10.7554/eLife.80880.sa2

## Additional files

### Supplementary files
• MDAR checklist

### Data availability

All code and data is available on Penn State ScholarSphere at this link: https://doi.org/10.26207/qpxv-5v60.

The following dataset was generated:

| Author(s) | Year | Dataset title | Dataset URL | Database and Identifier |
|---|---|---|---|---|
| Cellini M, Mongeau JM | 2022 | Unraveling nested feedback loops in insect gaze stabilization (Data for manuscript) | https://doi.org/10.26207/qpxv-5v60 | Scholarsphere, 10.26207/qpxv-5v60 |

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
