## [Editor Report]

The manuscript makes an important contribution to feedback control in neural systems. The analysis and modeling together make a compelling case for a nested system, combining visual with mechanosensory feedback, for head and body control in the fruit fly. The experiments that support these results are compelling and well-executed and the strategies for dissecting and modeling feedback are valuable to the field, and broadly applicable to other neural control systems. This paper will reach a wide audience; researchers investigating biological control systems, visual feedback, and gaze stabilization will all be interested in these results.

---

## [Decision Letter]

**Decision letter after peer review:**

Thank you for submitting your article "Unraveling nested feedback loops in insect gaze stabilization: Mechanosensory feedback actively damps visually guided head movements in fly flight" for consideration by *eLife*. Your article has been reviewed by 3 peer reviewers, one of whom is a member of our Board of Reviewing Editors, and the evaluation has been overseen by Ronald Calabrese as the Senior Editor. The reviewers have opted to remain anonymous.

All of the reviewers enjoyed this paper but thought that some revisions to the presentation of results and the discussion would improve the manuscript. New experiments are discussed in the individual reviews but are not required for this revision. However, the reviewers all felt that including alternate paradigms, hypotheses, and the experiments that do or could distinguish them are crucial adds to the text. Those and other essential elements are summarized here, with individual reviews included at the end:

Essential revisions:

1) It is important to distinguish this model from prior ones in flies, from ones in vertebrates, and from other potential models that could account for the data. This kind of hypothesis testing of model architectures seems like it would add a lot to the paper, especially if you could rule out classes of models and suggest multiple alternative models consistent with your data (and other data in the field). Please see R3's comments along these lines, especially.

2) Issues with the presentation of the results:

(2a) Presentation issues should be addressed to clarify experiments and what each is doing/testing. Reviewers found some of the figures hard to follow, which was surprising given what seemed like relatively straightforward modeling. Please see R2's comments along these lines, in particular.

(2b) All reviewers found the presentation of the nested versus feedback architecture confusing, on different levels. Definitely clarify if dissecting this is an assertion from the outset (and if so, please modify that claim according to the detailed feedback from R3 and R3), or a hypothesis that is being tested. If the latter, please make it easier to read out the weight of the evidence supporting the nested feedback hypothesis, along the lines of R1's comments.

*Reviewer #1 (Recommendations for the authors):*

This paper aims to dissect the structure of feedback control in the stabilization of gaze when both the external world and the body and head are in motion. To achieve this, sensory-motor systems must integrate visual and self-motion cues, but the precise structure of that integration is not generally known in invertebrate systems. The authors focus on the fly as a model system, where previous work establishes a firm grounding for the results but gaps in knowledge of how canonical experimental manipulations, e.g. anchoring the body, affect motor responses still abound. Using an elegant experimental design where the same visual inputs are delivered during body-fixed and body-free tethered flight, the authors are able to quantify how gaze stabilization is impacted by the two forms of feedback. The work reveals that visual feedback shifts the scale of head movements when the external world moves at different frequencies, but that the self-motion cues from body rotations serve to dampen head movements and are nested within the visual control feedback loop. The nonlinearity in this nested control system is quantified convincingly in the paper.

Main strengths:

– The experimental design and analyses are well-motivated and executed.

– There are clear differences between the head movements and frequencies responses to external visual perturbations in the head-fixed and head-free conditions.

– The proposed model accounts for the empirical data in the two scenarios nicely.

Main weaknesses:

– The strength of the evidence for the differentiation between the two feedback schemes was not clear, and Figures 4 and 5 were hard to follow without more information.

– It was not clear if the model proposed is unique as opposed to simply sufficient for explaining the empirical data.

The work will be of interest to motor and systems neuroscientists who study feedback control, across a broad range of species. Biomechanics researchers will benefit from the framework laid out here and this will inspire future work to uncover the possible mechanisms of this control. Beyond biology, engineers and robotics researchers will take interest in this kind of nested feedback control, for the design of bio-inspired robotic systems.

There is a strong assumption about the analytical form of the feedback gain control (G/(1+G)), and this needs a sentence at least of justification and background in the Results.

Figures 4 and 5 highlight the main results of the work, but it was hard to figure out the strength of the evidence for the nested control topology from the figures. It would greatly enhance the broader impact of the work if these figures were made more intuitive for the reader. Perhaps the figures could start by showing a cartoon of what the results should look like in the extreme case of each feedback scenario and weighting, to set expectations.

Are there other options for the control system that would produce different results in the body-fixed versus body-free flies? It seems like this isn't the only feedback control scheme possible, so a more careful discussion of why the one proposed might be the *unique* solution to the problem and match the data is crucial.

Something needs to be said in the Discussion about how this adds to what we already knew from the primate literature about nested VOR feedback within OKR feedback. Does this new work point to new mechanisms? In the OKR, there's been good work showing that similar feedback is achieved in primates and zebrafish, but with very different circuitry. Can similarly crisp claims be highlighted here?

Are there new experiments suggested by these results in other species that could broaden the impact of work in the future?

*Reviewer #2 (Recommendations for the authors):*

In this work, the authors present a model for mechanosensory feedback nested inside a visual feedback loop, both controlling body and head yaw rotations. Using a variety of experiments, they fit this model to behavioral data in the fruit fly, where head and body yaw rotations can be easily measured, and in some cases, feedback can be manipulated. They use this data to fit their model and draw conclusions about how different feedback loops interact to stabilize the gaze in the fly.

The strength of this paper is in its rigorous approach to modeling the feedback in the fly's interactions with the visual world. It manages to fit its model non-parametrically at several different ethologically relevant frequencies of feedback. The comparisons of behavior with and without mechanosensory feedback are illuminating, as is the comparison of voluntary with involuntary mechanical feedback. One weakness of the paper is in its presentation, which can be a little opaque for non-specialists in control theory.

This paper provides a methodology for dissecting how different feedback systems interact and combine to jointly control behavior. While the specific manipulations available in the fly are not universally available, the approach seems likely to be useful for investigating many systems.

Overall, this work looks well done and contributes valuably to understanding how head and body feedback systems work in tandem to stabilize gaze in flies. Most of my major comments relate to the presentation.

Major comments

1) In the introduction, it would help if the authors laid out a little more about what's known and not known, and what precisely this paper is adding to the literature. For instance, the authors state that it's already known that mechanosensory feedback represents nested feedback inside the visual feedback loop. So what's left is merely fitting the model to data? Or are there alternative models that could be tested and ruled out with this data? (If there are, I think the framework of testing alternatives could be powerfully convincing about how predictive this particular model is.) At the end of the introduction, I was left puzzled about what the authors were adding.

2) The stimulus pattern should be defined. Pictures show a square wave grating; is this accurate? Does it matter? What was the wavelength? It looks like a 30 d period or so from the illustrations, which would put maximum temporal frequencies of the moving pattern at ~250 d/s / (30 d) = 8 Hz, which is about right for maximally driving optomotor responses.

Questions:

a. The perturbation signal R is a displacement but is measured presumably as a velocity by the eyes, and the direction-selective signal from the eye is a nonlinear function of velocity. If the tuning of the velocity signal is different for guiding body vs. head movements, does that matter or does that fit easily into this theory? In the presented model, there's only one single visual feedback signal to both body and head.

b. In the fastest oscillating stimuli, the pattern only moves back and forth by 2 pixels or so, and I believe these LEDs have something like 8 brightness levels. Is the intended stimulus really accurately captured by this display?

3) The model section of the methods should be clearer about what the different signals and coefficients are. As I understand it, everything is complex, so the products represent both gain and phase shifts of sinusoids, represented as complex numbers. It would be helpful to define why R should be thought of as displacement rather than velocity, and whether H, B, and G represent angles or angular velocities. Head angle is relative to the body, so angle seems reasonable, but I'd expect body orientation signals to be angular velocities or even accelerations. This might all not matter since it's all in a linear framework, but I think this could nonetheless be made clearer to non-specialists by defining the variables and terminology more explicitly. In the text, there's a reference to a complex s, which I assume is part of the integrand for a Laplace transform, but this could be spelled out more clearly or not mentioned at all since Laplace transforms are otherwise avoided. Then these gain and phase shifts are computed for each frequency of the stimulus, and non-parametric curves are found for each complex coefficient.

4) There's at least one alternative way to break the feedback here, and I'm curious about why it wasn't used to test or fit models. Instead of breaking the mechanosensory feedback loop, one could leave it in place, and instead, place flies in a virtual open loop, so that there is no visual feedback from the behaviors. It might be hard to track the head in real time to do this, but I'm interested to know if there are tests of the theory that could result from this sort of perturbation to the system. Along the same lines, gluing the head to the thorax would remove one source of gaze feedback and could be used to test the model for body movements. Are these interesting tests to do? (I'm not necessarily asking for these experiments.)

*Reviewer #3 (Recommendations for the authors):*

The goal of this paper is to use the fruit fly *Drosophila melanogaster* to assess the relative contributions of vision and mechanosensory feedback in controlling head motion about the vertical, or yaw, axis. The authors perform a set of behavioral experiments comparing flies that are free to rotate in the yaw plane with rigidly tethered flies, using a control theoretic framework to make quantitative predictions about the importance of each sensory modality. They propose a model where mechanosensory feedback is nonlinearly integrated with visual feedback to control head steering, but only in the presence of whole-body rotations.

Overall, I find the paper well-written and the data very nicely presented. I appreciate the authors' formal use of control theory to make algebraic predictions about how the flies should respond to each perturbation and think this work adds a great deal to understanding the differences between free and tethered flight. I also like the conceptual approach of comparing parallel and nested sensory fusion problems in locomotion. That being said, I do have some major concerns about the approach that needs to be seriously addressed.

Control model and "eliminating" haltere feedback

This paper compares gaze stabilization in flies that can freely rotate about the yaw axis with those that are rigidly tethered. Crucially, in figure 2A, haltere feedback is presented as being a nested feedback loop that is only the result of the animal's body mechanics. In addition, the legend for 2C states, "Note that contributions of body visual and mechanosensory feedback are no longer present and all nested feedback is gone." In light of recent work, specifically Dickerson et al. 2019, I do not think the authors' view on either matter is correct. As that paper shows, the haltere is providing constant input to the wing steering system-even in the absence of body rotations (It is also worth noting that Fayazzuddin and Dickinson 1999 proposed a model of wing steering muscle function where the wing and haltere provide constant, rhythmic input). Those experiments relied on imaging from the haltere axon terminals in the brain that likely synapse onto neck motor neurons that help control gaze (Strausfeld and Seyan 1989). Moreover, that feedback is partially under visual control; the haltere steering muscles change the trajectory of the haltere in the presence of visual input alone, modulating the feedback it provides to the wing steering system. I am not sure if that makes the haltere system parallel or nested with the visual system, but it certainly means that haltere feedback is not solely due to body mechanics. More importantly, this knowledge of physiology means that in a rigidly tethered fly, the authors cannot fully eliminate haltere input. This has tremendous implications for their modeling efforts, as they can never fully bring Ghead,M to zero. This may explain why, in Figure 4, body visual feedback alone cannot account for changes in head gain. It also means that a diagram like Figure 5B is essentially not possible in an intact fly, as the haltere signal is ever-present.

Proposed neural architecture

The authors propose a model of head stabilization in which the visual system sends motor commands to the neck in parallel with a gating command to the haltere that is only present during body motion. To me, this is essentially the "control-loop" hypothesis, proposed by Chan et al. 1998 and confirmed by Dickerson et al. 2019. In that model, the halteres provide continuous, wingbeat-synchronous feedback during flight. As the fly takes visual input, the haltere steering muscle motor neurons receive commands relayed by the visual system, altering the haltere's motion. This, in turn, recruits more campaniform sensilla for each wing stroke, which fire at different preferred phases from those providing the initial rhythm signal. Then, due to the haltere's direct, excitatory connection with the wing steering muscles, this changes the timing or recruitment of the wing steering system, changing aerodynamic forces and the fly's trajectory. This suggests that the haltere's gyroscopic sensing is an epiphenomenon that coopts its likely ancestral role in regulating the timing of the wing steering system, rather than the other way around. Again, whether this means that the visual → haltere connection is parallel or nested within the visual loop proposed by the authors, I am not certain, though I lean toward the former. Additionally, it is crucial to note that the haltere has collateral projections to the neck motor centers. Thus, as the visual system manipulates haltere kinematics and mechanosensory feedback, the haltere is controlling head motion in a reciprocal fashion, even when there are no imposed body motions. Even the nonlinear gating of neck motor neurons the authors note here is not entirely in keeping with the model proposed by Huston and Krapp 2009. There, the presence of haltere beating or visual stimulus alone was not enough to cause the neck MNs to fire. However, simultaneous haltere beating and visual stimulus did, implying that the fly need only be in flight (or walking, in the case of Calliphora) for the halteres to help control head motion; Coriolis forces due to body rotations imposed or otherwise, need not be present. The only difference I can see between what the authors propose and the control-loop hypothesis is that they focus on the head (which, again, is covered by the revised model of Dickerson et al.) and that the nonlinear damping gate requires body motion (which is inconsistent with the findings of Huston and Krapp).

I think the most critical change is rethinking the control model of visual and mechanosensory feedback in light of our understanding of the haltere motor system. As noted earlier, the experiments with rigidly tethered flies do not fully eliminate haltere feedback, which greatly impacts the math used to make predictions about how the animals respond to various perturbations. I recognize this requires a severe overhaul of the manuscript, but my concern is that by considering the haltere as merely a passive gyroscopic sensor leaves out a number of potential explanations for the data in Figures 4 and 5. Additionally, the authors need to think hard about whether the haltere is controlled in parallel or nested with the visual system, given that they have a reciprocal relationship even in the case of a rigidly tethered fly.

I was rather surprised in the section about active damping of head saccades that there was almost no mention of the recent work by Kim et al. 2017 showing that head motion during saccades seems to follow a feedforward motor program (or Strausfeld and Seyan's 1988 (?) work detailing how vision and haltere info combine to help control head motion). Furthermore, the head velocities for body-free and rigidly tethered flies seem similar, which points to it being a feedforward motor program, a la Kim et al. If you subtract body displacement from the free-rotating head motion, do you get a similar result? That would hint that head isn't overcompensating during body-fixed experiments and is driven more reflexively, as proposed in the discussion. I would also recommend looking at Bartussek and Lehmann 2017 for the impact of haltere mechanosensory input on 'visuomotor' gain, or the work from the Fox lab.

Finally, the authors either need to detail how their model is distinct from the control-loop hypothesis or back off their claim of novelty and show that their work lends further evidence to that model. I would also prefer if the figure panel for the model is either more anatomically accurate or stuck with the block diagram framing of information flow.

---

## [Author Response]

Reviewer #1 (Recommendations for the authors):[…]There is a strong assumption about the analytical form of the feedback gain control (G/(1+G)), and this needs a sentence at least of justification and background in the Results.

Because the system we consider in this manuscript has nested feedback, the G/(1+G) form does not fully describe our model structure (as this only applies to the traditional one-sensor control system block diagram). If one were to consider G as the transform from sensory error E to head motion H (which contains the nested feedback loop in our framework), then we can use this expression to describe our model. In this case, the assumption is about the visual feedback, which we assume has a gain of -1. We believe this is a valid assumption because optic flow is inversely proportional to the motion of the eyes relative to the world. We describe in more detail why we believe this model structure is appropriate below. We have also added a line clarifying this idea when we introduce our model.

Figures 4 and 5 highlight the main results of the work, but it was hard to figure out the strength of the evidence for the nested control topology from the figures. It would greatly enhance the broader impact of the work if these figures were made more intuitive for the reader. Perhaps the figures could start by showing a cartoon of what the results should look like in the extreme case of each feedback scenario and weighting, to set expectations.

We agree that Figure 4–5 could benefit from being made more intuitive and have made multiple changes to make the presentation clearer.

For Figure 4, the baseline feedback case where there is only head visual feedback (purple curve) is what we would expect the data to look like in every experiment/prediction if body visual and mechanosensory feedback had no effect. Thus, we now show all the head data in Figure 4 with respect to this curve, which allows for more explicit comparisons. We have also updated the legend and now use cartoons to illustrate the effect of the different types of feedback.

For Figure 5D, we have added the baseline prediction for the ratio of the Ghead,V+M/Ghead,V if body mechanosensory feedback had no effect (gain = 1, phase = 0), and explicitly indicated that values <1 mean that head motion is damped by mechanosensory feedback.

Are there other options for the control system that would produce different results in the body-fixed versus body-free flies? It seems like this isn't the only feedback control scheme possible, so a more careful discussion of why the one proposed might be the unique solution to the problem and match the data is crucial.

In short, yes. A number of models with increasingly complex structures could be fit to our empirical data. However, we assert a parsimonious model structure that is well-suited for mathematically teasing apart the roles of the various sensory modalities during fly gaze stabilization. We believe that there is strong evidence supporting of our proposed model structure.

The match between our prediction of the effects of body visual feedback on the head response and the experimental replay experiment data (see Figure 4C, tan vs gray) is the most compelling evidence that supports our proposed model of visual feedback, as well as implying linearity. In a way, this is expected because the gain of the visual feedback loop (-1) is based on physics, i.e., moving one way elicits optic flow in an equal and opposite direction. The same visual feedback structure has been applied to model visuomotor tasks in other animals such as fish and moths, and has been shown to be similarly linear (Roth et al., 2011; Sponberg et al., 2015). To our knowledge, there has not been other models proposed for visual feedback in flies or other animals that would provide a more parsimonious explanation for the data.

In comparison to the structure of visual feedback, how flies integrate mechanosensory feedback during self-motion is slightly less clear. However, the *nested* structure of mechanosensory feedback is an inherent property of visually elicited behaviors—meaning that a visual stimulus elicits movement (due to the gaze stabilization reflex), which only then activates mechanosensory feedback, thus mechanosensory feedback is nested in this context. For clarity, the nested feedback architecture is not a hypothesis, but an assertion, when locomotion is reflexively driven by visual inputs. Our model is broadly consistent with previous work which suggested an inner-loop (nested) structure for haltere feedback (Elzinga et al., 2012). The model structure is also consistent with the role of antenna feedback in damping groundspeed control in flies (Fuller et al., 2014).

How mechanosensory is combined with visual information in the brain is not fully resolved. Seminal work showed that there is feedback from the halteres to the flight control muscles, even in the absence of visual inputs (Dickinson, 1999; Fayyazuddin and Dickinson, 1996). Furthermore, visual and haltere inputs due to body motion sum when presented together in rigidly tethered flies, although wing responses (like head responses we measured) were somewhat uncoordinated about the yaw axis (Sherman and Dickinson, 2004, 2003). A limitation of these prior studies is that they considered haltere inputs that were externally generated and in open-loop (flies were rigidly tethered to a motor) so it is difficult to say whether the same topology applies to gyroscopic haltere inputs due to self-motion, i.e. nested feedback. However, these data from prior studies do support the summing of visual and mechanosensory inputs at the neural level, which we maintain in our model structure (as outlined in Figure 2A). A major contribution of our work is the synthesis of a parsimonious linear model that can capture the empirical data.

For transparency, the parallel control topology we show in Figure 1A is not a permissible model structure for our study because it explicitly assumes that both sensory systems (i.e., visual and mechanosensory) receive the same external reference input. The general form of this model (based on Figure 1A) is:Y(s)=PCS1+PCS21+PCS1+PCS2R(s). In this case, even if one of the sensory systems is abolished (for example vision in flies):Y(s)=PCS21+PCS2R(s), there will be a nonzero response Y(s) to the refence input R(s). This parallel model is excellent for examining a fly’s response to coupled visual and haltere inputs (e.g., due to a gust of wind, etc.), but it is not appropriate for our analysis because the fly’s motion is reflexively driven by visual inputs. We present the parallel model to distinguish the nested model structure from prior models, e.g., (Roth et al., 2016).

In the experimental paradigm we employed, there is no external reference for the halteres, thus any feedback must be nested (meaning that there will not be any haltere feedback due to body motion unless visual motion first elicits body motion). This nested model takes the form (based on Figure 2B):

Y(s)=PCS11+PCS1−PCS2R(s), where abolishing S1 leads to R(s), no matter what the input R(s) is. We now explicitly state from the onset that the nested feedback architecture we employ is an assertion, not a hypothesis.

Something needs to be said in the Discussion about how this adds to what we already knew from the primate literature about nested VOR feedback within OKR feedback. Does this new work point to new mechanisms? In the OKR, there's been good work showing that similar feedback is achieved in primates and zebrafish, but with very different circuitry. Can similarly crisp claims be highlighted here?

To our knowledge, by and large the primate literature on the VOR/OKR considers these visual and mechanosensory feedback loops summing in a parallel topology (due to the experimental design)—as opposed to mechanosensory signals being nested with visual feedback (reviewed in (Goldberg et al., 2012)). Further, to our knowledge, only a select few studies in primates have considered nested mechanosensory feedback (Schweigart et al., 1997), but have not attempted to unravel the contributions of the different feedback modalities in shaping the control of the head/eyes in the way that we have here. We now briefly discuss these ideas in the introduction and have added a few sentences in the discussion outlining the new contributions of our work.

Are there new experiments suggested by these results in other species that could broaden the impact of work in the future?

Yes, there are absolutely some interesting experiments focused on uncovering the role of nested mechanosensory/proprioceptive feedback that could be carried out in other species. Indeed, most animals have proprioceptive sensory systems that likely support higher level behaviors. In insects without halteres, such as moths, the antennae are thought to fulfill this role (Sane et al., 2007). We have added a Discussion section titled “Nested proprioception across phyla” considering these ideas.

Reviewer #2 (Recommendations for the authors):[…]Overall, this work looks well done and contributes valuably to understanding how head and body feedback systems work in tandem to stabilize gaze in flies. Most of my major comments relate to the presentation.Major comments1) In the introduction, it would help if the authors laid out a little more about what's known and not known, and what precisely this paper is adding to the literature. For instance, the authors state that it's already known that mechanosensory feedback represents nested feedback inside the visual feedback loop. So what's left is merely fitting the model to data? Or are there alternative models that could be tested and ruled out with this data? (If there are, I think the framework of testing alternatives could be powerfully convincing about how predictive this particular model is.) At the end of the introduction, I was left puzzled about what the authors were adding.

While it is generally accepted that mechanosensory feedback due to self-motion (i.e., nested feedback) is present during visual driven tasks, the structure of this feedback and how it interacts with visual feedback during gaze stabilization in flight is presently unclear. Our work is the first (to our knowledge) to propose a neuromechanical model for the integration of visual and nested mechanosensory feedback based on empirical data and to quantify the effects of nested feedback on gaze stabilization. We agree that the introduction could benefit from clarification and have added a few sentences in the text discussing the novelty of our work.

2) The stimulus pattern should be defined. Pictures show a square wave grating; is this accurate? Does it matter? What was the wavelength? It looks like a 30 d period or so from the illustrations, which would put maximum temporal frequencies of the moving pattern at ~250 d/s / (30 d) = 8 Hz, which is about right for maximally driving optomotor responses.

Yes, we displayed a square wave pattern with a spatial wavelength of 30°, which is accurately illustrated in Figure 2. This is specified in the methods section titled “Visual perturbations”. We now clarify in this section that this spatial wavelength with our prescribed visual motion yields mean temporal frequencies of ~5 Hz (with a max of ~8 Hz), which is right around the optimum of the motion vision pathway in *Drosophila* (Duistermars et al., 2007a).

Questions:a. The perturbation signal R is a displacement but is measured presumably as a velocity by the eyes, and the direction-selective signal from the eye is a nonlinear function of velocity. If the tuning of the velocity signal is different for guiding body vs. head movements, does that matter or does that fit easily into this theory? In the presented model, there's only one single visual feedback signal to both body and head.

The tuning of the velocity signal is in fact different for guiding the head and body, but this fits quite nicely into our theory. The tuning of the head and body with respect to visual motion is accounted for in their corresponding neural controllers (Chead,V and Cbody,V) shown in Figure 2A. Effectively, Chead,V and Cbody,V represent how the brain processes R differently for the head and body. We have shown in our previous work (Cellini et al., 2022) that Cbody,V is tuned closely to velocity while Chead,V is actually tuned more strongly to acceleration. This difference in tuning is what explains why the body has a low-pass filter response and the head is more like a high-pass filter, as shown in Figure 2A,C and Figure 3A-B, corroborating our recent work (Cellini et al., 2022).

A nice property of our model and non-parametric approach is that is does not matter what the baseline tuning of the head and body are (velocity sensitive, acceleration sensitive, etc.). We isolate and measure Ghead,V and Gbody,V (which contain Chead,V and Cbody,V) by removing mechanosensory feedback in body-fixed flies, and then use the response of body-free flies to work backwards and compute the effects of mechanosensory feedback. This way, our model can account for the differences in tuning between body-free and body-fixed flies, rather than focusing on the tuning itself. We have added a sentence in the text clarifying that Chead,V and Cbody,V account for differences in tuning between the head and body.

b. In the fastest oscillating stimuli, the pattern only moves back and forth by 2 pixels or so, and I believe these LEDs have something like 8 brightness levels. Is the intended stimulus really accurately captured by this display?

The fastest oscillating stimuli does indeed only move between three pixels (one at the 0 position, and then 3.75°m in each direction). This results in something similar to a square wave being displayed on our flight simulator (see Author response image 1, left). However, when transformed into frequency-domain, the resulting signal is very similar to the idealized one (see Author response image 1, right). We account for any discrepancies in our analysis by using the actual displayed signal as the input to our model. This ensures that we are modeling a fly’s response to the frequency content of what they are actually seeing. We would not expect a fly’s response to be much different between the idealized and actual signals anyway, because a fly’s visual system low-pass filters high-frequency components of visual motion, which would make the square wave appear smoother than it actually is, e.g. see (Duistermars et al., 2007b) for a confirmation experiment. We now clarify in the text that we record the voltage signal from our visual display, which measures the displacement of our stimulus, and use that signal as the input to our model.

**Author response image 1. sa2fig1:** *Left*: the idealized smooth sine wave at the highest frequency designed for our flight simulator (black) vs the actual displayed signal (red). Note that our flight simulator display has an angular resolution of 3.75°. *Right*: same as the left, but for the Fast-Fourier Transform (FFT) magnitude of the two signals.

3) The model section of the methods should be clearer about what the different signals and coefficients are. As I understand it, everything is complex, so the products represent both gain and phase shifts of sinusoids, represented as complex numbers. It would be helpful to define why R should be thought of as displacement rather than velocity, and whether H, B, and G represent angles or angular velocities. Head angle is relative to the body, so angle seems reasonable, but I'd expect body orientation signals to be angular velocities or even accelerations. This might all not matter since it's all in a linear framework, but I think this could nonetheless be made clearer to non-specialists by defining the variables and terminology more explicitly. In the text, there's a reference to a complex s, which I assume is part of the integrand for a Laplace transform, but this could be spelled out more clearly or not mentioned at all since Laplace transforms are otherwise avoided. Then these gain and phase shifts are computed for each frequency of the stimulus, and non-parametric curves are found for each complex coefficient.

We fully acknowledge that flies primarily measure the velocity of wide-field visual perturbations with their visual system, as this has been shown extensively in prior work. Therefore, the gain and phase between the body  B and visual perturbation R can be most precisely thought of as a ratio of velocities in frequency domain. However, a nice mathematical property of linear frequency-domain system identification is that this ratio is equivalent for displacements, velocities, accelerations, etc. For example, consider the case where if B and R are defined as displacements. Multiplying them by the complex variable s is equivalent to taking their derivatives in frequency domain (i.e., converting them to velocities) so the ratio is then defined as: sBsR=BR The s term cancels out, so our results would not be affected by converted these signals to velocities, or simply keeping them as displacements. However, computing displacement from velocity involves taking a numerical derivative, which amplifies noise (Van Breugel et al., 2020), thus we prefer to make all calculations using displacements. To make these points clearer, we have clarified in the text the definition of the complex variable s and explained our reasoning for making our calculations using displacements instead of velocities.

4) There's at least one alternative way to break the feedback here, and I'm curious about why it wasn't used to test or fit models. Instead of breaking the mechanosensory feedback loop, one could leave it in place, and instead, place flies in a virtual open loop, so that there is no visual feedback from the behaviors. It might be hard to track the head in real time to do this, but I'm interested to know if there are tests of the theory that could result from this sort of perturbation to the system. Along the same lines, gluing the head to the thorax would remove one source of gaze feedback and could be used to test the model for body movements. Are these interesting tests to do? (I'm not necessarily asking for these experiments.)

We find this approach and the corresponding experiments tremendously intriguing, as it involves more potential manipulations to the control topology that could provide insight into the inner workings of the feedback system. One could potentially use a virtual/augmented reality system to abolish visual feedback from the head and/or body in real-time, which would correspond to changing the model from Equation 3 (now Equation 5) in the text to any of the three following forms:

1) Removed body visual feedback: H=Ghead,V1+Ghead,VR +Ghead,M1+Ghead,VB

2) Removed head visual feedback: H=Ghead,V(R−B)+Ghead,MB

3) Removed head and body visual feedback: H=Ghead,VR+Ghead,MB

These expressions make predictions for the corresponding experiments in the virtual reality system and could provide further validation of our model. Our group has just recently developed a system to accomplish real-time virtual reality for the body in the magnetic tether system—although not yet for the head, as this is substantially more challenging to do in real-time than offline due to the need to find the neck joint in each frame for a rotating body. Thus, only the feedback case described by the equation in (1) is possible for us to achieve experimentally at present. (1) corresponds to Equation 6 (now Equation 8) in the text, and the prediction we made about the head response when body visual feedback is removed but mechanosensory feedback is still present (Figure 4D, cyan). Based on this prediction we would expect the head to operate with lower gain than when the body is fixed, but higher gain than when the body is free with natural body visual feedback. We are currently working on another manuscript that is beyond the scope of this study and now briefly discuss how these feedback manipulations fit into our framework within the discussion.

Regarding the second point about fixing the head and how this might affect the control of the body, we have previously published data on these experiments (for the sum-of-sines visual inputs). The data are presented in our previous paper (Cellini et al., 2022), but not integrated with the framework we introduce here. Our previous analysis showed that fixing the head has a modest, but significant, effect on the body gain and phase at high frequencies (where the head is typically the most active), which can be predicted by our control model. While these results support our proposed model and are further evidence for linearity, we believe that including these data/modeling is a bit beyond the scope of the current manuscript and does not directly address the role of nested mechanosensory feedback. Therefore, we prefer to leave this out of the manuscript.

Reviewer #3 (Recommendations for the authors):The goal of this paper is to use the fruit fly *Drosophila melanogaster* to assess the relative contributions of vision and mechanosensory feedback in controlling head motion about the vertical, or yaw, axis. The authors perform a set of behavioral experiments comparing flies that are free to rotate in the yaw plane with rigidly tethered flies, using a control theoretic framework to make quantitative predictions about the importance of each sensory modality. They propose a model where mechanosensory feedback is nonlinearly integrated with visual feedback to control head steering, but only in the presence of whole-body rotations.Overall, I find the paper well-written and the data very nicely presented. I appreciate the authors' formal use of control theory to make algebraic predictions about how the flies should respond to each perturbation and think this work adds a great deal to understanding the differences between free and tethered flight. I also like the conceptual approach of comparing parallel and nested sensory fusion problems in locomotion. That being said, I do have some major concerns about the approach that needs to be seriously addressed.Control model and "eliminating" haltere feedbackThis paper compares gaze stabilization in flies that can freely rotate about the yaw axis with those that are rigidly tethered. Crucially, in figure 2A, haltere feedback is presented as being a nested feedback loop that is only the result of the animal's body mechanics. In addition, the legend for 2C states, "Note that contributions of body visual and mechanosensory feedback are no longer present and all nested feedback is gone." In light of recent work, specifically Dickerson et al. 2019, I do not think the authors' view on either matter is correct. As that paper shows, the haltere is providing constant input to the wing steering system-even in the absence of body rotations (It is also worth noting that Fayazzuddin and Dickinson 1999 proposed a model of wing steering muscle function where the wing and haltere provide constant, rhythmic input). Those experiments relied on imaging from the haltere axon terminals in the brain that likely synapse onto neck motor neurons that help control gaze (Strausfeld and Seyan 1989). Moreover, that feedback is partially under visual control; the haltere steering muscles change the trajectory of the haltere in the presence of visual input alone, modulating the feedback it provides to the wing steering system. I am not sure if that makes the haltere system parallel or nested with the visual system, but it certainly means that haltere feedback is not solely due to body mechanics. More importantly, this knowledge of physiology means that in a rigidly tethered fly, the authors cannot fully eliminate haltere input. This has tremendous implications for their modeling efforts, as they can never fully bring Ghead,M to zero. This may explain why, in Figure 4, body visual feedback alone cannot account for changes in head gain. It also means that a diagram like Figure 5B is essentially not possible in an intact fly, as the haltere signal is ever-present.

We thank the reviewer for the detailed insights into the neurobiology of the haltere-wing system. We agree that our phrasing and terminology regarding “eliminating haltere feedback” could be misleading and needs to be revised.

As the reviewer points out, the halteres have been implicated in two primary functions:

1) as a gyroscope to sense body motion and

2) as a ‘metronome’ that regulates and structures the timing of motor outputs via tonic input (wings, and likely the head as well).

In our work, we have focused exclusively on function (1), as the emphasis of this manuscript is on how body motion influences head movements (from both vision and proprioception). Therefore, when we state that we have eliminated haltere (or mechanosensory) feedback, we are referring to eliminating function (1) of the halteres, i.e. inputs due to body motion.

As the reviewer accurately states, we have not eliminated all the functions of the halteres by fixing the body. This could only be achieved by ablating the halteres themselves (or potentially with genetic silencing), which would result in both functions (1) and (2) being eliminated. While bilaterally removing the halteres is an option in body-fixed flies and has been shown to have a modest effect on head movements (Mureli et al., 2017), this is unfortunately not possible in body-free flies (free or magnetically tethered) because flies immediately become unstable. This means that, even if we were to ablate the halteres in body-fixed flies, it would be difficult to infer if any differences in head responses we observed between body-free and body-fixed flies were due to functions (1) or (2). Keeping the halteres intact in both cases ensures that function (2) is active in both cases and any differences we observe between body-free and body-fixed flies are due to function (1).

Fortunately, even though we cannot remove function (2) of the halteres, we believe our model is still appropriate for understanding the role of function (1) without major alterations to the framework. As the reviewer states, visual motion drives steering commands of the head/wings in parallel with commands to the halteres themselves, regardless of whether body motion occurs. These haltere commands structure the timing of motor output in conjunction with visual inputs. Therefore, we think of function (2) more as a sub-component of the visual controller Ghead,V that mediates optomotor steering than as part of the mechanosensory controller Ghead,M that senses body motion (although haltere movements certainly affect how gyroscopic information is measured). Thus, we can lump function (2) in with the visual controller, rather than the mechanosensory controller, in order to investigate the role of function (1) of the halteres—because function (1) is the only thing that changes between conditions. For clarity, when we fix the body to isolate and measure Ghead,V, function (2) is still present, and thus contained within the dynamics of Ghead,V. This same Ghead,V is active in body-free flies; thus, we can be sure that any differences we see between body-free and body-fixed flies in not due to function (2). While we acknowledge that this means we have not truly isolated the pure visual controller (because function 2) is always present, this does not affect any of our conclusions about function (1) of the halteres (i.e., gyroscopic inputs due to body motion).

To reconcile our model with prior work on function (2) of the halteres, we have added a section in the beginning of the results discussing the dual-function role of the halteres and changed our language throughout to be clear that fixing the body only removes function (1). Specifically, we now specify “…eliminated haltere/mechanosneosry feedback due to body motion…”, to distinguish from haltere feedback due to function (2).

Proposed neural architectureThe authors propose a model of head stabilization in which the visual system sends motor commands to the neck in parallel with a gating command to the haltere that is only present during body motion. To me, this is essentially the "control-loop" hypothesis, proposed by Chan et al. 1998 and confirmed by Dickerson et al. 2019. In that model, the halteres provide continuous, wingbeat-synchronous feedback during flight. As the fly takes visual input, the haltere steering muscle motor neurons receive commands relayed by the visual system, altering the haltere's motion. This, in turn, recruits more campaniform sensilla for each wing stroke, which fire at different preferred phases from those providing the initial rhythm signal. Then, due to the haltere's direct, excitatory connection with the wing steering muscles, this changes the timing or recruitment of the wing steering system, changing aerodynamic forces and the fly's trajectory. This suggests that the haltere's gyroscopic sensing is an epiphenomenon that coopts its likely ancestral role in regulating the timing of the wing steering system, rather than the other way around. Again, whether this means that the visual → haltere connection is parallel or nested within the visual loop proposed by the authors, I am not certain, though I lean toward the former. Additionally, it is crucial to note that the haltere has collateral projections to the neck motor centers. Thus, as the visual system manipulates haltere kinematics and mechanosensory feedback, the haltere is controlling head motion in a reciprocal fashion, even when there are no imposed body motions. Even the nonlinear gating of neck motor neurons the authors note here is not entirely in keeping with the model proposed by Huston and Krapp 2009. There, the presence of haltere beating or visual stimulus alone was not enough to cause the neck MNs to fire. However, simultaneous haltere beating and visual stimulus did, implying that the fly need only be in flight (or walking, in the case of Calliphora) for the halteres to help control head motion; Coriolis forces due to body rotations imposed or otherwise, need not be present. The only difference I can see between what the authors propose and the control-loop hypothesis is that they focus on the head (which, again, is covered by the revised model of Dickerson et al.) and that the nonlinear damping gate requires body motion (which is inconsistent with the findings of Huston and Krapp).

We further thank the reviewer for these insights. It is our understanding that the “control-loop” model proposed by (Chan et al., 1998) primarily refers to the halteres’ role in structuring motor output. This model is well supported by (Dickerson et al., 2019), but does not yet reveal how gyroscopic inputs due to body motion might be modulated by haltere muscles (as all experiments were performed on body-fixed flies and the specific campaniform arrays involved in sensing gyroscopic forces are currently ambiguous). However, the discussion in (Dickerson et al., 2019) sets up some nice hypotheses for how haltere muscles might act to recruit haltere campaniforms of different types—some that are *insensitive* to gyroscopic forces, and some that are *sensitive* to gyroscopic forces. We believe our work builds on these hypotheses and provides preliminary evidence that gyroscopic-sensitive inputs might be actively modulated by visual inputs, as flies’ head responses were strikingly different when they were controlling their own body motion *vs* when body motion was externally imposed. While our data does not provide evidence at the level of neural circuits, we believe our work stands nicely next to (Dickerson et al., 2019) and (Chan et al., 1998) and provides additional hypotheses to be tested. We now include additional discussion in the text (Discussion section: Distinguishing between self-generated and externally generated body motion) on these ideas and reconcile our data with prior models in the literature.

I think the most critical change is rethinking the control model of visual and mechanosensory feedback in light of our understanding of the haltere motor system. As noted earlier, the experiments with rigidly tethered flies do not fully eliminate haltere feedback, which greatly impacts the math used to make predictions about how the animals respond to various perturbations. I recognize this requires a severe overhaul of the manuscript, but my concern is that by considering the haltere as merely a passive gyroscopic sensor leaves out a number of potential explanations for the data in Figures 4 and 5. Additionally, the authors need to think hard about whether the haltere is controlled in parallel or nested with the visual system, given that they have a reciprocal relationship even in the case of a rigidly tethered fly.

See our response above for more detail. From our understanding of the metronome function of the halteres, i.e., providing tonic input, this internal feedback loop should be equally present in both body-free and body-fixed flies, and therefore does not offer a clear justification for why we see damping of head movements in body-free, but not body-fixed, flies. We have clarified in the introduction and the beginning of the results that, while the halteres serve to structure motor output independently of body motion, that any differences we observe in body-free flies strongly suggest that haltere inputs due to body motion (gyroscope forces) are the underlying cause**.**

I was rather surprised in the section about active damping of head saccades that there was almost no mention of the recent work by Kim et al. 2017 showing that head motion during saccades seems to follow a feedforward motor program (or Strausfeld and Seyan's 1988 (?) work detailing how vision and haltere info combine to help control head motion). Furthermore, the head velocities for body-free and rigidly tethered flies seem similar, which points to it being a feedforward motor program, a la Kim et al. If you subtract body displacement from the free-rotating head motion, do you get a similar result? That would hint that head isn't overcompensating during body-fixed experiments and is driven more reflexively, as proposed in the discussion. I would also recommend looking at Bartussek and Lehmann 2017 for the impact of haltere mechanosensory input on 'visuomotor' gain, or the work from the Fox lab.

Kim et al., 2017 argues that head roll during a head saccade follows a feedforward motor program based on data showing that the head will roll (in addition to yaw) to offset body roll during a saccade, even though there is no body roll in the magnetic tether they used for experiments. We confirmed this result in a recent paper and provided further evidence for the feedforward hypothesis by studying head saccades in a rigid tether where there is no body motion (Cellini et al., 2021). While we agree that this data is consistent with a *visually* open-loop feedforward motor program, we believe that our data strongly supports the idea that mechanosensory feedback is present during saccades, especially during braking, which is consistent with previous work in a similar paradigm (Bender and Dickinson, 2006). While the peak velocities of head saccades in body-free and body-fixed flies are similar in our data (although statistically different), the most prominent difference is in how long it takes the head to return to baseline after the initial rapid movement (as shown in Figure 7). Furthermore, the amplitude of head saccades in body-free flies is considerably smaller than in body-fixed flies. These stark differences, even in the absence of visual features (see Figure S7), strongly suggest that mechanosensory feedback from body motion underlies this behavior. This is consistent with Kim et al. 2017, as our data still show that head saccades are likely visually open loop (visual features don’t change response). The rapid (5-10ms) response time of mechanosensory feedback is also well within the saccade duration (~50ms), so it is reasonable to assume that mechanosensory information can shape head saccade dynamics, even if they are visually open-loop. All head saccade data are presented in the body reference frame (head relative to the body), so subtracting (or adding) body movement would not change this interpretation. We have clarified in the text that head saccades are visually open-loop, but that mechanosensory feedback likely mediates braking. We now also cite (Kim et al., 2017), (Milde et al., 1987) , and (Strausfeld and Seyan, 1985) in the section discussing saccades. Note that we believe the reviewer was referring to Strausfeld and Seyan's 1985 and/or 1987 work (Milde et al., 1987; Strausfeld and Seyan, 1985), as we could not find a relevant study from 1988.

While (Bartussek and Lehmann, 2016; Lehmann and Bartussek, 2017) and (Kathman and Fox, 2019; Mureli et al., 2017; Mureli and Fox, 2015) present intriguing results describing how local haltere/wing proprioception shapes motor output (likely by modulating visuomotor gain), we feel that because these studies focus on the tonic function of the haltere (all experiments in body-fixed flies), that they do not address the role of the *gyroscopic* haltere inputs we investigate here. We have added a line in the Results section on saccades clarifying these ideas.

Finally, the authors either need to detail how their model is distinct from the control-loop hypothesis or back off their claim of novelty and show that their work lends further evidence to that model. I would also prefer if the figure panel for the model is either more anatomically accurate or stuck with the block diagram framing of information flow.

See our above comments. We have also modified the panel in Figure 7E to follow our block diagram format.

References

Bartussek J, Lehmann F-O. 2016. Proprioceptive feedback determines visuomotor gain in *Drosophila*. *R Soc Open Sci* 3. doi:10.1098/rsos.150562

Bender JA, Dickinson MH. 2006. A comparison of visual and haltere-mediated feedback in the control of body saccades in *Drosophila melanogaster*. *J Exp Biol* 209:4597–4606. doi:10.1242/jeb.02583

Cellini B, Mongeau J-M. 2020. Active vision shapes and coordinates flight motor responses in flies. *Proc Natl Acad Sci* 117:23085–23095. doi:10.1073/pnas.1920846117

Cellini B, Salem W, Mongeau J-M. 2021. Mechanisms of punctuated vision in fly flight. *Curr Biol* 31:4009-4024.e3. doi:10.1016/j.cub.2021.06.080

Cellini B, Salem W, Mongeau J-MM. 2022. Complementary feedback control enables effective gaze stabilization in animals. *Proc Natl Acad Sci* 119:e2121660119. doi:https://doi.org/10.1073/pnas.2121660119

Chan WP, Prete F, Dickinson MH. 1998. Visual Input to the Efferent Control System of a Fly’s “Gyroscope.” *Science (80- )* 280:289–292. doi:10.1126/science.280.5361.289

Dickerson BH, de Souza AM, Huda A, Dickinson MH. 2019. Flies Regulate Wing Motion via Active Control of a Dual-Function Gyroscope. *Curr Biol* 29:3517-3524.e3. doi:10.1016/j.cub.2019.08.065

Dickinson MH. 1999. Haltere–mediated equilibrium reflexes of the fruit fly, *Drosophila melanogaster*. *Philos Trans R Soc London Ser B Biol Sci* 354:903–916. doi:10.1098/rstb.1999.0442

Duistermars BJ, Chow DM, Condro M, Frye MA. 2007a. The spatial, temporal and contrast properties of expansion and rotation flight optomotor responses in *Drosophila*. *J Exp Biol* 210:3218–3227. doi:10.1242/jeb.007807

Duistermars BJ, Reiser MB, Zhu Y, Frye MA. 2007b. Dynamic properties of large-field and small-field optomotor flight responses in *Drosophila*. *J Comp Physiol A Neuroethol Sensory, Neural, Behav Physiol* 193:787–799. doi:10.1007/s00359-007-0233-y

Elzinga MJ, Dickson WB, Dickinson MH. 2012. The influence of sensory delay on the yaw dynamics of a flapping insect. *J R Soc Interface* 9:1685–1696. doi:10.1098/rsif.2011.0699

Fayyazuddin A, Dickinson MH. 1996. Haltere Afferents Provide Direct, Electrotonic Input to a Steering Motor Neuron in the Blowfly, Calliphora. *J Neurosci* 16:5225–5232. doi:10.1523/JNEUROSCI.16-16-05225.1996

Fuller SB, Straw AD, Peek MY, Murray RM, Dickinson MH. 2014. Flying *Drosophila* stabilize their vision-based velocity controller by sensing wind with their antennae. *Proc Natl Acad Sci* 111:E1182–E1191. doi:10.1073/pnas.1323529111

Goldberg JM, Wilson VJ, Cullen KE, Angelaki DE, Broussard DM, Buttner-Ennever J, Fukushima K, Minor LB. 2012. The Vestibular System, The Vestibular System: A Sixth Sense. Oxford University Press. doi:10.1093/acprof:oso/9780195167085.001.0001

Heisenberg M, Wolf R. 1986. Vision in *Drosophila*. Genetics in Microbehavior. Studies of Brain Function, Volume 12. M. Heisenberg , R. Wolf. *Q Rev Biol* 61:141–141. doi:10.1086/414849

Kathman ND, Fox JL. 2019. Representation of Haltere Oscillations and Integration with Visual Inputs in the Fly Central Complex. *J Neurosci* 39:4100–4112. doi:10.1523/JNEUROSCI.1707-18.2019

Kim AJ, Fenk LM, Lyu C, Maimon G. 2017. Quantitative Predictions Orchestrate Visual Signaling in *Drosophila*. *Cell* 168:280-294.e12. doi:10.1016/j.cell.2016.12.005

Lehmann F-O, Bartussek J. 2017. Neural control and precision of flight muscle activation in *Drosophila*. *J Comp Physiol A* 203:1–14. doi:10.1007/s00359-016-1133-9

Milde JJ, Seyan HS, Strausfeld NJ. 1987. The neck motor system of the fly Calliphora erythrocephala – II. Sensory organization. *J Comp Physiol A* 160:225–238. doi:10.1007/BF00609728

Mureli S, Fox JL. 2015. Haltere mechanosensory influence on tethered flight behavior in *Drosophila*. *J Exp Biol* 218:2528–2537. doi:10.1242/jeb.121863

Mureli S, Thanigaivelan I, Schaffer ML, Fox JL. 2017. Cross-modal influence of mechanosensory input on gaze responses to visual motion in *Drosophila*. *J Exp Biol* 220:2218–2227. doi:10.1242/jeb.146282

Roth E, Hall RW, Daniel TL, Sponberg S. 2016. Integration of parallel mechanosensory and visual pathways resolved through sensory conflict. *Proc Natl Acad Sci* 113:12832–12837. doi:10.1073/pnas.1522419113

Roth E, Sponberg S, Cowan N. 2014. A comparative approach to closed-loop computation. *Curr Opin Neurobiol* 25:54–62. doi:10.1016/j.conb.2013.11.005

Roth E, Zhuang K, Stamper SA, Fortune ES, Cowan NJ. 2011. Stimulus predictability mediates a switch in locomotor smooth pursuit performance for Eigenmannia virescens. *J Exp Biol* 214:1170–1180. doi:10.1242/jeb.048124

Sane SP, Dieudonné A, Willis MA, Daniel TL. 2007. Antennal Mechanosensors Mediate Flight Control in Moths. *Science (80- )* 315:863–866. doi:10.1126/science.1133598

Schweigart G, Mergner T, Evdokimidis I, Morand S, Becker W. 1997. Gaze Stabilization by Optokinetic Reflex (OKR) and Vestibulo-ocular Reflex (VOR) During Active Head Rotation in Man. *Vision Res* 37:1643–1652. doi:10.1016/S0042-6989(96)00315-X

Sherman A, Dickinson MH. 2004. Summation of visual and mechanosensory feedback in *Drosophila* flight control. *J Exp Biol* 207:133–142. doi:10.1242/jeb.00731

Sherman A, Dickinson MH. 2003. A comparison of visual and haltere-mediated equilibrium reflexes in the fruit fly *Drosophila melanogaster*. *J Exp Biol* 206:295–302. doi:10.1242/jeb.00075

Sponberg S, Dyhr JP, Hall RW, Daniel TL. 2015. Luminance-dependent visual processing enables moth flight in low light. *Science (80- )* 348:1245–1248. doi:10.1126/science.aaa3042

Stöckl AL, Kihlström K, Chandler S, Sponberg S. 2017. Comparative system identification of flower tracking performance in three hawkmoth species reveals adaptations for dim light vision. *Philos Trans R Soc B Biol Sci* 372:20160078. doi:10.1098/rstb.2016.0078

Strausfeld NJ, Seyan HS. 1985. Convergence of visual, haltere, and prosternai inputs at neck motor neurons of Calliphora erythrocephala. *Cell Tissue Res* 240:601–615. doi:10.1007/BF00216350

Van Breugel F Van, Kutz JN, Brunton BW. 2020. Numerical Differentiation of Noisy Data: A Unifying Multi-Objective Optimization Framework. *IEEE Access* 8:196865–196877. doi:10.1109/ACCESS.2020.3034077

Windsor SP, Taylor GK. 2017. Head movements quadruple the range of speeds encoded by the insect motion vision system in hawkmoths. *Proc R Soc B Biol Sci* 284:20171622. doi:10.1098/rspb.2017.1622

Wolf R, Voss A, Hein S, Heisenberg M, Sullivan GD. 1992. Can a fly ride a bicycle? *Philos Trans R Soc London Ser B Biol Sci* 337:261–269. doi:10.1098/rstb.1992.0104